# Long-term trends and drivers of aerosol pH in eastern China
Min Zhou[1,2], Guangjie Zheng[3], Hongli Wang[1], Liping Qiao[1], Shuhui Zhu[1], DanDan Huang[1], Jingyu An[1],
Shengrong Lou[1], Shikang Tao[1], Qian Wang[1], Rusha Yan[1], Yingge Ma[1], Changhong Chen[1], Yafang Cheng[3],
Hang Su[*,1,4], Cheng Huang[1]
[1]State Environmental Protection Key Laboratory of the Cause and Prevention of Urban Air Pollution
Complex, Shanghai Academy of Environmental Sciences, Shanghai200233, China
[2]School of Atmospheric Sciences, Nanjing University, Nanjing210023, China
[3]Minerva Research Group, Max Planck Institute for Chemistry, Mainz 55128, Germany
[4]Multiphase Chemistry Department, Max Planck Institute for Chemistry, Mainz 55128, Germany
*Corresponding author: Hang Su (h.su@mpic.de)

**20** **Abstract**

**21**     Aerosol acidity plays a key role in regulating the chemistry and toxicity of atmospheric aerosol particles.

**22**     The trend of aerosol pH and its drivers are crucial in understanding the multiphase formation pathways

**23**     of aerosols. Here, we reported the first trend analysis of aerosol pH from 2011 to 2019 in eastern China,

**24**     calculated with the ISORROPIA model based on observed gas and aerosol compositions. The

**25**     implementation of the Air Pollution Prevention and Control Action Plan led to -35.8%, -37.6%, -9.6%, -

**26**     81.0% and 1.2% changes of $PM_{2.5}$, $SO_4^{2-}$, $NH_x$, non-volatile cations (NVCs) and $NO_3^-$ in the Yangtze

**27**     River Delta (YRD) region during this period. Different from the drastic changes of aerosol compositions

**28**     due to the implementation of the Air Pollution Prevention and Control Action Plan, aerosol pH showed

**29**     a minor change of -0.24 over the 9 years. Besides the multiphase buffer effect, the opposite effects from

**30**     the changes of $SO_4^{2-}$ and non-volatile cations played key roles in determining this minor pH trend,

**31**     contributing to a change of +0.38 and −0.35, respectively. Seasonal variations in aerosol pH were mainly

**32**     driven by the temperature, while the diurnal variations were driven by both temperature and relative

**33**     humidity. In the future, $SO_2$, $NO_x$ and $NH_3$ emissions are expected to be further reduced by 86.9%, 74.9%

**34**     and 41.7% in 2050 according to the best health effect pollution control scenario (SSP1-26-BHE). The

**35**     corresponding aerosol pH in eastern China is estimated to increase by ~0.19, resulting in 0.04 less $NO_3^-$

**36**     and 0.12 less $NH_4^+$ partitioning ratios, which suggests that $NH_3$ and $NO_x$ emission controls are effective

**37**     in mitigating haze pollution in eastern China.

**38**

**39**     **1 Introduction**

**40**     Aerosol acidity is an important parameter in atmospheric chemistry. It affects the particle mass and

**41**     chemical composition by regulating the reactions of aerosols, and is closely associated with human health,

**42**     ecosystems and climate (Li et al., 2017; Nenes et al., 2020b; Pye et al., 2020; Su et al., 2020). Aerosol

**43**     acidity has attracted an increasing concern in recent years because of its impacts on the thermodynamics

**44**     of gas-particle partitioning, pH-dependent condensed-phase reactions and trace metal solubility (Cheng

**45**     et al., 2016; Fang et al., 2017; Guo et al., 2017b; Guo et al., 2016; He et al., 2018; Song et al., 2018;

**46**     Weber et al., 2016; Su et al., 2020; Tilgner et al., 2021).

Thermodynamic models, such as E-AIM (Clegg et al., 1998) and ISORROPIA II, are commonly used
for aerosol pH estimations, due to the limitations and difficulties in direct measurements (Fountoukis and
Nenes, 2007; Hennigan et al., 2015). Previously reported aerosol pH generally ranged from -1 to 6 on a
global scale (Pye et al., 2020; Zheng et al., 2020; Su et al., 2020). In the United States, aerosols were
reported to be highly acidic, with pH values of approximately 0–2 (Guo et al., 2015; Nah et al., 2018;
Pye et al., 2018; Zheng et al., 2020). In comparison, aerosols in mainland China and Europe were
generally less acidic with aerosol pH ranging between 2.5 and 6 (Guo et al., 2018; Jia et al., 2018; Masiol
et al., 2020; Shi et al., 2019; Tan et al., 2018; Wang et al., 2019; Zheng et al., 2020).
Aerosol pH exhibits notable spatial and temporal variabilities due to changes in factors such as
temperature, relative humidity (RH), and aerosol compositions (Pye et al., 2018; Nenes et al., 2020a; Tao
et al., 2020; Zheng et al., 2020). Very few studies have investigated the trend and spatial variability of
aerosol pH and its drivers. Weber et al. (2016) showed that aerosols remained highly acidic upon large
(~70%) reduction of particulate sulfate ($SO_4^{2-}$) during summertime in the southeastern United States over
the past 15 years. Based on the 10-year observations conducted at six Canadian sites, Tao and Murphy
(2019) suggested that meteorological parameters were more important than the chemical compositions
in controlling aerosol pH. Zheng et al. (2020) found that aerosol liquid water content (ALWC) and
temperature were the main factors that contribute to the pH difference observed between the wintertime
North China Plain and summertime southeastern United States, whereas the change of chemical
composition only played a minor role (15%). In China, the long-term trend of aerosol pH and its drivers
remain poorly understood, especially in recent years when the emissions and aerosol compositions
changed substantially.
To tackle severe particulate matter pollution in China, the Chinese government released the Air
Pollution Prevention and Control Action Plan (hereinafter referred to as the Action Plan) in September
2013, which is the first plan specifying air quality goals in China (Cai et al., 2017; Liu et al., 2018; Zheng
et al., 2018). The implementation of the Action Plan has led to significant changes in the concentrations
and chemical compositions of fine particulate matter ($PM_{2.5}$), thus may altering aerosol pH and
subsequently feedback to the multiphase formation pathways of aerosols such as sulfate, nitrate and
ammonium (Cheng et al., 2016; Vasilakos et al., 2018; Nenes et al., 2020a).
In this study, we performed a comprehensive analysis on the long-term trends of aerosol pH and its
drivers in Shanghai, China. The thermodynamic model ISORROPIA II (version 2.1) (Fountoukis and
Nenes, 2007) was applied to estimate the pH based on 9-year continuous online measurements of $PM_{2.5}$
compositions at an urban site in Shanghai. The main purposes of this study are to: (1) characterize the
long-term trend of aerosol pH; (2) investigate the seasonal and diurnal variations of aerosol pH and the
main factors that affect these changes and (3) predict future pH under different emission control scenarios.
The results presented here can help advance our understanding in aerosol chemistry, providing a scientific
basis to the development of effective pollution control strategy in the future.
**2 Material and Methods**
**2.1 Ambient measurements**
The observation site in this study is located at the Shanghai Academy of Environmental Sciences (SAES,
31°10′N, 121°25′E), which sits in the densely populated city centre of Shanghai (Figure S1). In the
absence of significant nearby industrial sources, this sampling site represents a typical urban area of
Shanghai affected by emissions from vehicular traffic, commercial, and residential activities (Qiao et al.,
2014; Zhou et al., 2016).
Gases and $PM_{2.5}$ components were continuously sampled by an on-line analyser to monitor aerosols
and gases (MARGA ADI 2080, Applikon Analytical B.V) from 2011 to 2019. Hourly mass
concentrations of major inorganic components were obtained, including gaseous components, i.e.,
hydrogen chloride (HCl), nitrous acid ($HNO_2$), sulfur dioxide ($SO_2$), nitric acid ($HNO_3$), ammonia ($NH_3$)
and particulate components, i.e., $SO_4^{2-}$, nitrate ($NO_3^-$), chloride ($Cl^-$), ammonium ($NH_4^+$), sodium ($Na^+$),
potassium ($K^+$), calcium ($Ca^{2+}$) and magnesium ($Mg^{2+}$). Details of measurements have been given in
Qiao et al. (2014), thus are only briefly described here. To better track the changes in retention time of
different ion species and ensure their concentrations were measured correctly, an internal standard check
was conducted every hour with lithium bromide (LiBr) standard solution (Qiao et al., 2014; Zhou et al.,
2016). The sampling system of MARGA was cleaned and multi-point calibrations with the standard
solutions were performed every three months to ensure the accuracy of measurements. To ensure the data
quality, ion balance between the measured charge equivalent concentrations of cation ($NH_4^+$, $Na^+$, $K^+$,
$Ca^{2+}$ and $Mg^{2+}$) and anion ($SO_4^{2-}$, $NO_3^-$ and $Cl^-$) species was examined as shown in Figure S2. Strong
correlation ($R^2$ = 0.94) was found between the cations and anions, suggesting good data quality during
the measurement period. We note that data during 2011-2016 were more scattered than those during
2017-2019, likely due to the significant decreases in $Ca^{2+}$, $K^+$ and $Mg^{2+}$ from 2011 to 2019 (Figure S3-
S5). In previous studies, intercomparison experiments between MARGA and filter-based method have
been carried out, and the data measured by MARGA showed acceptable accuracy and precision (Rumsey
et al., 2014; Huang et al., 2014; Stieger et al., 2018). A Thermal/Optical Carbon Aerosol Analyzer (model
RT-4, Sunset laboratory Inc.) equipped with a $PM_{2.5}$ cyclone was used for the organic carbon
measurement at a time resolution of 1 hour. The $PM_{2.5}$ mass concentrations were measured
simultaneously using an on-line beta attenuation PM monitor (FH 62 C14 series, Thermo Fisher
Scientific) at a time resolution of 5 min.

Temperature and RH, which are important factors affecting aerosol pH, were also measured at a time

resolution of 1 min. Annually averaged temperature and RH from 2011 to 2019 are shown in Figure S6.
The $t$-test results revealed that temperature rose significantly at a rate of 1.2 %/yr ($p < 0.01$), while RH
changed little.

**2.2 Aerosol pH prediction**
The aerosol pH was predicted using the ISORROPIA II thermodynamic model (Fountoukis and Nenes,
2007). ISORROPIA II can calculate the equilibrium $H_{air}^+$ and aerosol liquid water content of inorganic
material ($ALWC_i$) with the input of concentrations of the total $SO_4^{2-}$ ($TH_2SO_4$, replaced by observed
$SO_4^{2-}$), total $NO_3^-$ ($TNO_3$, gas $HNO_3$ plus particle $NO_3^-$), total ammonia ($NH_x$, gas $NH_3$ plus particle
$NH_4^+$), total $Cl^-$ ($TCl$, replaced by observed $Cl^-$ due to the low concentration and large measurement
uncertainties of HCl) (Fu et al., 2015; Ding et al., 2019), non-volatile cations (NVCs, observed $Na^+$, $K^+$,
$Ca^{2+}$, $Mg^{2+}$) and meteorological parameters (temperature and RH) (Guo et al., 2016). $H_{air}^+$ and $ALWC_i$
are then used to obtain the $PM_{2.5}$ pH by Eq. (1).
$$pH = -log_{10}H_{aq}^+ \cong -log_{10}\frac{1000H_{air}^+}{ALWC_i+ALWC_o} \cong -log_{10}\frac{1000H_{air}^+}{ALWC_i} \; , \tag{1}$$
where $H_{aq}^+$ is the $H^+$ concentration in solution (mol/L), $H_{air}^+$ is the $H^+$ loading for an air sample ($\mu g/m^3$)
and $ALWC_i$ and $ALWC_o$ are the aerosol liquid water contents of inorganic and organic species,
respectively ($\mu g/m^3$). $ALWC_o$ is calculated by Eq. (2) (Guo et al., 2015).
$$\text{ALWC}_o = \frac{m_{org}\rho_w}{\rho_{org}} \frac{\kappa_{org}}{(\frac{1}{RH}-1)} \ , \tag{2}$$
where $m_{org}$ is the mass concentration of organic aerosol, $\rho_w$ is the density of water ($\rho_w$=1.0g/cm$^3$),
$\rho_{org}$ is the density of organics ($\rho_{org}$=1.4g/cm$^3$) (Guo et al., 2015), and $\kappa_{org}$ is the hygroscopicity
parameter of organic aerosol ($\kappa_{org} = 0.087$) (Li et al., 2016). The concentration of organic aerosol was
estimated by multiplying the measured concentration of organic carbon by a factor of 1.6 (Turpin and
Lim, 2001). The average concentrations of $\text{ALWC}_o$ and $\text{ALWC}_i$ in Shanghai from 2011 to 2019 were
4.1 ($\pm$10.2) and 32.6 ($\pm$52.5) μg/m$^3$, respectively. $\text{ALWC}_o$ only accounted for 11.1% of the total aerosol
liquid water content. The pH predictions in previous studies were insensitive to $\text{ALWC}_o$ unless the
mass fraction of $\text{ALWC}_o$ to the total aerosol liquid water content was close to unity (Guo et al., 2015).
The use of $\text{ALWC}_i$ to predict pH is therefore fairly accurate and common (Battaglia et al., 2017; Ding
et al., 2019; Battaglia Jr et al., 2019). In this study, ISORROPIA II was run in the forward mode and
'metastable' state. Calculations using total (gas and aerosol) measurements in the forward mode are
less affected by measurement errors (Hennigan et al., 2015; Song et al., 2018). A detailed description of
the pH calculations can be found in previous studies (Guo et al., 2017a; Guo et al., 2015; Song et al.,

2018).

Figure S7 compares the predicted vs. measured concentrations of $NH_3$, $NH_4^+$, $NO_3^-$ and $HNO_3$. The
results show that the predicted and measured concentrations of $NH_3$, $NH_4^+$ and $NO_3^-$ are in good
agreements ($R^2$ values all over 0.89 and slopes close to 1.00), indicating that the thermodynamic analysis
accurately represents the aerosol state. However, the predicted and measured concentrations of $HNO_3$
are not well correlated, which is also observed in previous studies (Ding et al., 2019; Guo et al., 2015).
The reason for the gap can be attributed to (1) lower concentrations of gas-phase $HNO_3$ than that of
particle-phase $NO_3^-$, (2) MARGA has high uncertainty for $HNO_3$ measurement (Rumsey et al., 2014).
The development of an alternative approach is therefore warranted to accurately represent $HNO_3$ in the
future.
**2.3 Drivers of aerosol pH variations**
To investigate the factors that drive changes in aerosol pH, sensitivity tests of different factors on pH
variations, including temperature, RH, $SO_4^{2-}$, $TNO_3$, $NH_x$, $Cl^-$ and NVCs, were performed with the one-
at-a-time method. That is, assuming the aerosol pH estimated under scenario I (pH$_I$) differs from that
under scenario II (pH$_{II}$), the pH difference, (ΔpH = pH$_{II}$ – pH$_I$), are thus caused by the variations in the
factors listed above. To quantify the contributions of individual factors, we varied the factor $i$ from the
value in scenario I to the value in scenario II while keeping the other factors constant. The corresponding
changes in pH, ΔpH$_i$, were assumed to represent the contribution of the change of this individual factor
to the overall aerosol pH variations. Note that because of the nonlinear dependence of pH to different
factors, the sum of contributions of individual factors can be slightly different from the overall
contributions of all factors. The unresolved contributors to pH differences, i.e., $\Delta pH - \sum_i \Delta pH_i$, were
attributed to "others", which might represent the contribution of covariations between the factors. This
method was used for the results presented in Figure 1b, Figure 3 and Figure 5, where the corresponding
scenarios represented the average conditions in different years (Figure 1b), seasons (Figure 3) or diurnal
periods (Figure 5).
**3 Results and Discussion**
**3.1 Long-term trends of aerosol pH**
**3.1.1 Trends of aerosol pH**
The 9-year time series of aerosol pH calculated by ISORROPIA II is shown in Figure 1a. A declining
trend in PM$_{2.5}$ pH from 3.30 ± 0.58 in 2011 to 3.06 ± 0.55 in 2019 was observed, with the fitted decrease
rate of around 0.04 pH per year, which may be related to chemical composition changes (Figure S8-S9)
due to the pollution control measures taken in the Yangtze River Delta (YRD) region. The Chinese
government started to carry on the Action Plan, a series of air pollution control policies, in September
2013, which resulted in declines in PM$_{2.5}$ and its major components (Cheng et al., 2019; Li et al., 2019).
Compared to the concentrations before the implementation of the Action Plan (i.e., average of 2011-2012
averages), PM$_{2.5}$, SO$_4^{2-}$, NH$_x$ and NVCs during 2018-2019 decreased by 35.8%, 37.6%, 9.6% and 81.0%,
respectively, while NO$_3^-$ increased by 1.2% (Fig. S8). Through the years, SO$_4^{2-}$, NH$_4^+$ and NO$_3^-$ remained
the most abundant inorganic water-soluble ions, accounting for 83.4%–94.1% of the total ions in PM$_{2.5}$.
While the proportions of NH$_4^+$ and NO$_3^-$ showed continuous increases (increased by 2.2% and 13.1%
from 2011 to 2019, respectively), those of NVCs and SO$_4^{2-}$ decreased by 6.0% and 4.6%, respectively.
Despite the substantial changes of aerosol abundance and composition, the aerosol pH only showed a
minor change. The effects of changes in $PM_{2.5}$ chemical composition on the aerosol pH will be detailed
in Section 3.1.2.

The $PM_{2.5}$ in Shanghai was moderately acidic with a daily pH averaging 3.18 and ranging from 1.15

to 5.62, similar to those from other cities in China (Shi et al., 2019; Tan et al., 2018). Compared with
other countries globally (Table S1), aerosol pH values in Chinese cities of 1.82 to 5.70 were higher than
those in US cities of 0.55 to 2.20 (Guo et al., 2015; Pye et al., 2018; Nah et al., 2018), yet similar to those
in European cities of 2.30 to 3.90 (Guo et al., 2018; Masiol et al., 2020). Among all of the Chinese cities,
the aerosol pH was highest in Inner Mongolia, which might be caused by a higher contribution of crustal
dust (Wang et al., 2019). The pH values in Shanghai and Guangzhou were lower than those in North
China, which may be due to higher concentrations of $NH_3$ and dust emissions over the latter region (Shi
et al., 2007; Liu et al., 2019).
**1.1.2  Driving factors.**
Figure 1b shows the contributions of individual factors to the ∆pH from 2011 to 2019. Here the bar plots
indicate the factors contributing to the ∆pH between two adjacent scenarios as shown in Figure 1b, e.g.,
2011 and 2013. See Figure S10a for the factor contribution to the variation from average conditions. Note
that in Fig. 1b, the aerosol pH was calculated from the annual averages of input parameters. This is
different from Sect 3.1.1, where the annual pH was the average of hourly values based on hourly
observation data. As shown in Figure 1b, the aerosol pH decreased from 3.35 in 2011 to 3.28 in 2013.
The main factors that affected the pH during 2011-2013 (prior to the implementation of the Action Plan)
were the temperature and NVCs. The pH value also continuously decreased from 3.28 in 2013 to 3.19 in
2019. Yet, chemical composition showed more prominent effects on the aerosol pH during 2013-2019
compared to that of 2011-2013. As aforementioned, upon implementation of the Action Plan (2013-2019),
the concentrations of $PM_{2.5}$ and its chemical components decreased substantially (Figure S8). Changes
of $SO_4^{2-}$ and NVCs were important determinants in the change of aerosol pH, resulting in ∆pH of +0.38
and −0.35 respectively from 2013 to 2019. Changes in the $NH_x$ and $Cl^-$ contributed 0.08 and 0.06
decreases in ∆pH, respectively, whereas $TNO_3$ had little impact on the ∆pH. Hence, besides the effect of
reduction in $SO_4^{2-}$ (Fu et al., 2015; Xie et al., 2020), our results suggest that the change in NVCs may
also play an important role in determining the trend of aerosol pH. During 2017-2019, temperature and
$NH_x$ became the main drivers of the ∆pH. The effects of $SO_4^{2-}$ and NVCs on pH were much weaker than
those during 2013–2017, consistent with the fact that the declines in pollutant concentrations slowed
down in recent years (Fig. S9).
Overall, the changes in $SO_4^{2-}$ and NVCs were the main drivers of the ΔpH upon the implementation
of the Action Plan, and $NH_x$ appeared to play an increasingly important role in determining the aerosol
pH through the years.

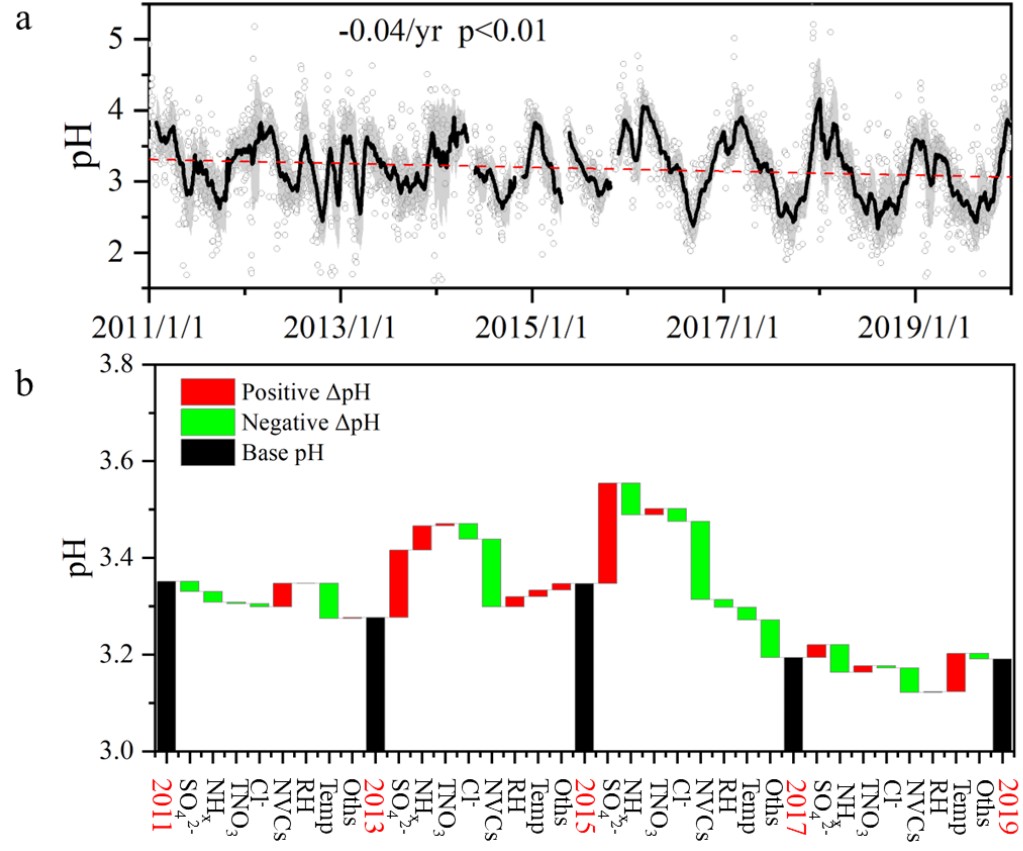


**Figure 1. (a) Long-term trends in aerosol pH during 2011–2019 in Shanghai.** Gray dots and black lines
represent the daily pH values and 30-day moving average pH values, respectively. Shaded areas mark the standard
deviation of 30-day moving average pH values. **(b) Contributions of individual factors to the ΔpH from 2011 to**
**2019.** Here the black bars indicate the mean pH of different years, and the red and green bars represent the positive
and negative effects of individual factors on ΔpH between two adjacent scenarios, e.g., 2011 and 2013,
respectively. The meanings of the abbreviations: RH, relative humidity; Temp, temperature; NVCs, non-volatile
cations; $NH_x$, total ammonia; $TNO_3$, total nitrate; Oths, others.

## 3.2 Seasonal variation

Figure 2 shows the seasonal variations of aerosol pH in Shanghai. The average pH values were $3.33\pm$ 0.49, $2.89 \pm 0.49$, $2.99 \pm 0.52$ and $3.59 \pm 0.57$ in spring (March–May, MAM), summer (June–August, JJA), fall (September–November, SON) and winter (December–February, DJF), respectively. The highest aerosol pH was found in winter while the lowest pH was found in summer. While the seasonal variations of pH in Shanghai were similar to those observed in Beijing and other cities in North China Plain (Tan et al., 2018; Ding et al., 2019; Shi et al., 2019; Wang et al., 2020), the absolute values were lower, due to the generally lower $PM_{2.5}$ concentrations in YRD.

Figure 3 shows the contributions of individual factors to the $\Delta$pH across the four seasons. Here the bar plots indicate the factors contributing to the $\Delta$pH between two adjacent seasons, e.g., spring (MAM) and summer (JJA). See Figure S10b for the factor contribution to the variation from average conditions. The aerosol pH was calculated from the mean averages of input parameters in four seasons, and the $\Delta$pH was estimated by varying one factor while holding the other factors fixed in different seasons. According to the multiphase buffer theory, the peak buffer pH, $pK_a^*$, regulates the aerosol pH in a multiphase-buffered system, and temperature can largely drive the seasonal variation of aerosol pH through its impact on $pK_a^*$ (Zheng et al., 2020). This is evidenced by the results in Figure 3, as temperature showed a dominant role in driving the seasonal variation of aerosol pH. The temperature was associated with a maximum $\Delta$pH of 0.63 from fall to winter. Besides temperature, other two main factors were $NH_x$ and $SO_4^{2-}$ (Figure 3), contributing 16% and 12% of the changes, respectively. Our results suggest a central role of temperature in the determination of seasonal variations in aerosol pH, consistent with the results of Tao and Murphy (Tao and Murphy, 2019) at six Canadian sites and the prediction by the multiphase buffer theory (Zheng et al., 2020). In comparison, some previous studies emphasized the importance of chemical compositions in seasonal variations (Tan et al., 2018; Ding et al., 2019), which is mainly due to the different sensitivity analysis methods applied.

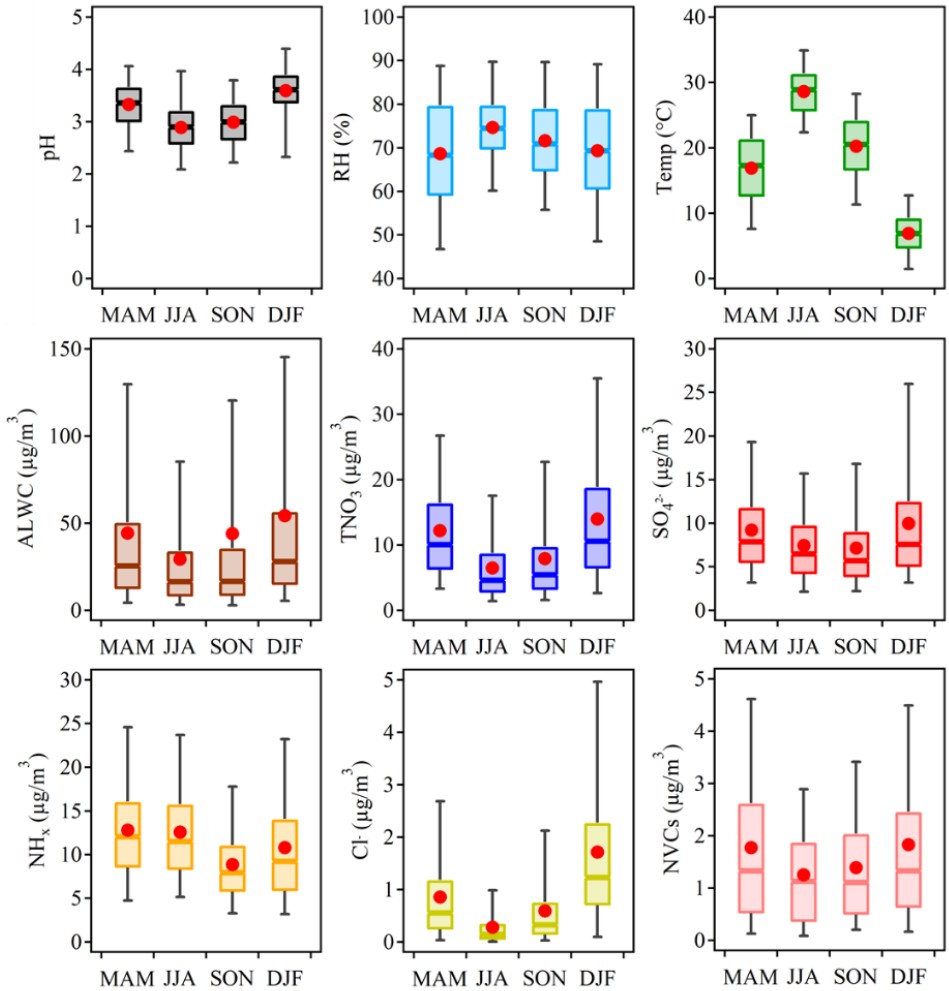

**Figure 2. Seasonal variations of the mass concentrations of major components in PM2.5, relative humidity**

**(RH), temperature (Temp), predicted aerosol liquid water content (ALWC) and aerosol pH during 2011–2019**

**in Shanghai.**

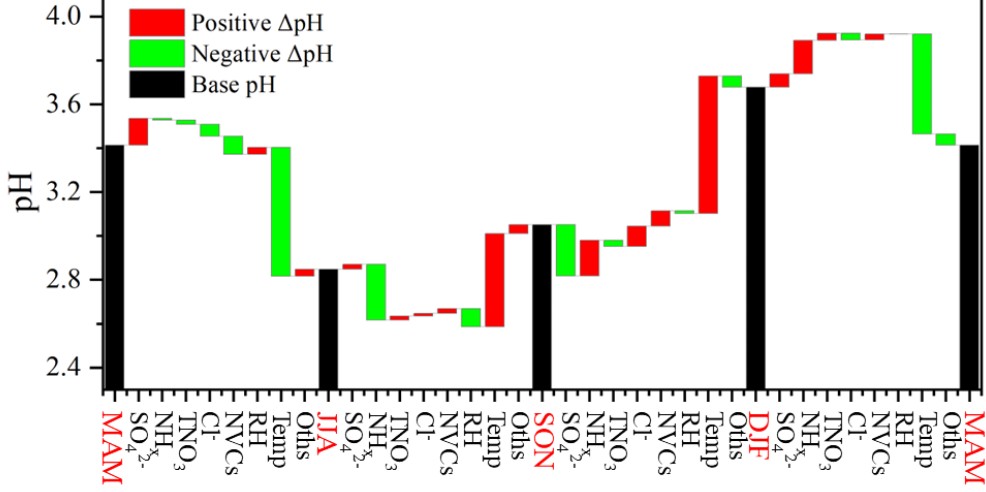

**Figure 3. Contributions of individual factors to the ΔpH across the four seasons.** Here the black bars indicate

the mean pH of different seasons, and the red and green bars represent the positive and negative effects of individual

factors on ΔpH between two adjacent scenarios, e.g., spring (MAM) and summer (JJA), respectively. The meanings

of the abbreviations: RH, relative humidity; Temp, temperature; NVCs, non-volatile cations; $NH_x$, total ammonia;

$TNO_3$, total nitrate; Oths, others.

**3.3 Diurnal variation**

Aerosol pH in Shanghai exhibited notable diurnal variations with higher aerosol acidity observed during

daytime. Diurnal variations of aerosol pH as well as those of its potential drivers are depicted in Figure

4. We further explored the effects of individual factors on the ΔpH between day and night through

sensitivity tests.

The bar plots in Figure 5 indicate the factors contributing to the ΔpH between two adjacent hour

periods, e.g., 0:00 and 6:00. See Figure S10c for the combined effects of contributions from different

factors on the average ΔpH. The aerosol pH was calculated from the averages of input parameters in 0:00,

6:00, 12:00 and 18:00, and ΔpH was estimated by varying one factor while holding the other factors

fixed in different hours. Temperature and RH were among the main drivers of the diurnal variation of

aerosol pH, with a maximum ΔpH of -0.22 and +0.10, respectively. As shown in Figure 4, the maximum

values of RH and ALWC occurred at approximately 5:00. After sunrise, the increase in temperature

resulted in an immediate drop of RH with ALWC reaching its lowest level in the afternoon. Accordingly,

the minimum aerosol pH (~2.8) was also found in the afternoon with high temperature and low RH. After

sunset, the decreasing temperature and increasing RH led to the highest aerosol pH overnight. Minor

changes in pH were found between 0:00 and 6:00, when temperature and RH also showed minor changes.

The impacts of other factors, such as $SO_4^{2-}$, on the diurnal variations of pH were notably smaller than

those on seasonal variations, which may be attributed to the relatively small variations of chemical

profiles during the course of a day. Among the chemical compositions, $NH_x$ played the most important

role, followed by $SO_4^{2-}$. Overall, temperature and RH were more important than chemical compositions

in regulating the diurnal variations of aerosol pH.

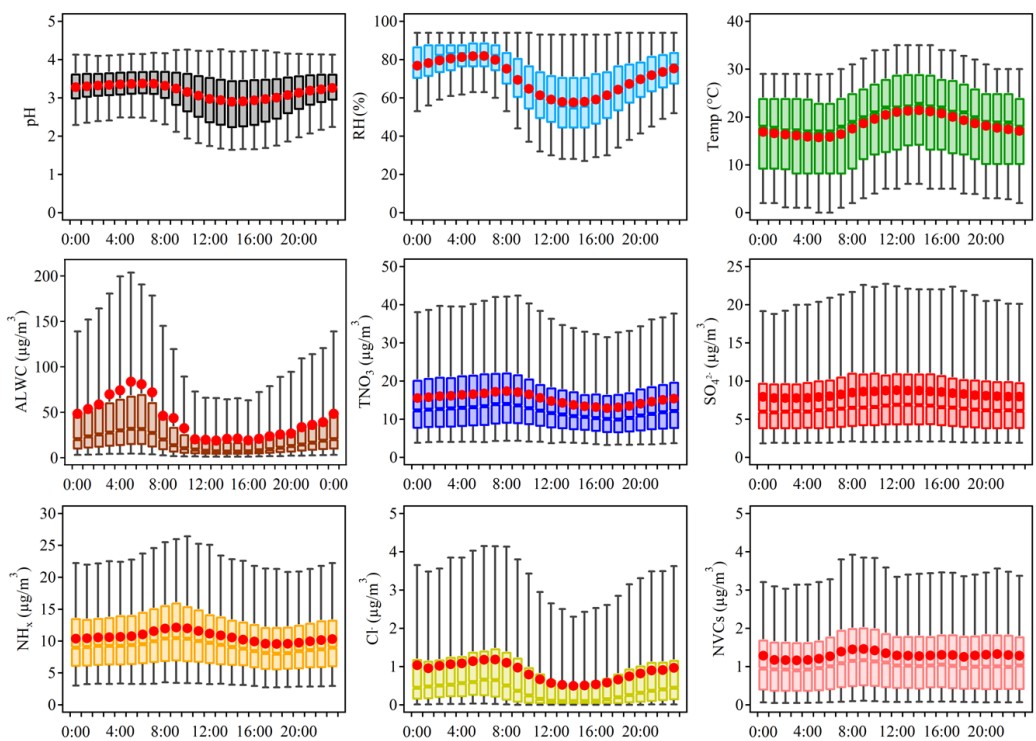


**Figure 4. Diurnal variations of the mass concentrations of major ions in PM$_{2.5}$, relative humidity (RH), temperature (Temp), predicted aerosol liquid water content (ALWC) and aerosol pH during 2011–2019 in Shanghai.**





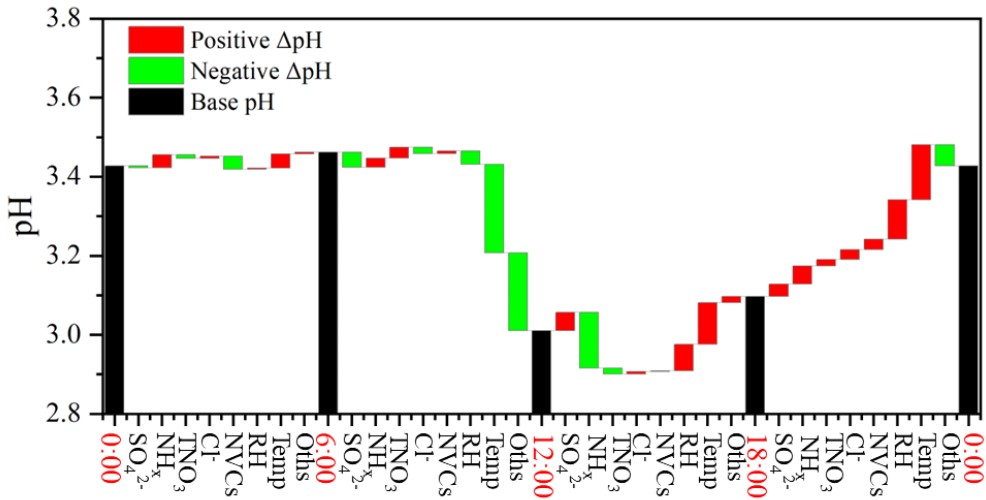


**Figure 5. Contributions of individual factors to the ΔpH between day and night.** Here the black bars indicate the mean pH of different hours, and the red and green bars represent the positive and negative effects of individual factors on ΔpH between two adjacent scenarios, e.g., 0:00 and 6:00, respectively. The meanings of the abbreviations:




RH, relative humidity; Temp, temperature; NVCs, non-volatile cations; $NH_x$, total ammonia; $TNO_3$, total nitrate;
Oths, others.

**3.4 Future projections**

A series of prevention and control measures have been suggested for continuous improvement in air
quality, which are expected to affect particulate compositions and subsequently alter aerosol pH in China.
To explore China's future anthropogenic emission pathways in 2015–2050, Tong et al. (2020) developed
a dynamic projection model, based on which different emission scenarios were created by connecting
five socio-economic pathway (SSP) scenarios, five representative concentration pathways (RCP)
scenarios (RCP8.5, 7.0, 6.0, 4.5 and 2.6) and three pollution control scenarios (business as usual, BAU;
enhanced control policy, ECP; and best health effect, BHE). These scenarios provide a better
understanding of future trends in pollutant emissions (Tong et al., 2020).
In this study, we chose three different emission reduction scenarios (SSP3-70-BAU, SSP2-45-ECP,
and SSP1-26-BHE) as the future anthropogenic emission pathways, and based on which we tried to
project future aerosol pH levels in Shanghai. SSP1-26-BHE, which involves a combination of strong
low-carbon and air pollution control policy, has the greatest emission reduction, followed by SSP2-45-
ECP. SSP3-70-BAU is a reference scenario without additional efforts to constrain emissions. We first
tested the sensitivity of aerosol abundances to precursor emissions with the historical data (Figure S11),
where the emissions of Shanghai were obtained by the Multi-resolution Emission Inventory for China
(MEIC, http://meicmodel.org/, last access: 15 January 2020). We found that the non-volatile sulfate
concentrations generally correlated linearly with that of the $SO_2$ emissions. For the volatile $TNO_3$ and
$NH_x$, the correlations are less linear, likely due to the different deposition velocities of gases and particles
(Pye et al., 2020; Weber et al., 2016; Nenes et al., 2021). The historical emission reductions have resulted
in a moderate pH decrease (Figure 1), a moderate increase (0.2% per year) in the $NO_3^-$ partitioning, and
a decrease (-0.6% per year) in the $NH_4^+$ partitioning (Figure S12).
For a first-order estimation, we applied the average $\Delta$ aerosol / $\Delta$ (precursor emissions) in ($\mu g/m^3$)
/ (Gg/yr) as derived from the historical data (Figure S11a-c) to the future scenario predictions. Figure 6
shows the projected emissions of $SO_2$, $NO_x$, $NH_3$, the predicted pH levels, and the effects of major
chemical components ($NH_x$, $SO_4^{2-}$, and $TNO_3$) to the $\Delta pH$ in Shanghai from 2015 to 2050 under the three
scenarios. Based on this assumption, the concentrations of $SO_4^{2-}$, $NO_3^-$ and $NH_4^+$ are expected to drop to
~6.3, 5.7 and 2.6 μg/m$^3$, respectively, in 2050 with the SSP1-26-BHE scenario, generally in agreement
with the predicted $PM_{2.5}$ levels of ~15 μg/m$^3$ under a similar scenario (Shi et al., 2021).
Under the reference scenario of SSP3-70-BAU with weak control policy (blue dashed lines in Figure
6a-f), $SO_2$ and $NO_x$ are predicted to increase, while the $NH_x$ is relatively stable. $NH_x$, $SO_4^{2-}$, and $TNO_3$
have minor effects on ΔpH (Figure 6g). Correspondingly, there are little changes in aerosol pH and the
predicted $NO_3^-$ partitioning ratio ($NO_3^-$ / ($NO_3^-$ + $HNO_3$)). However, the $NH_4^+$ partitioning ratio ($NH_4^+$ /
($NH_4^+$ + $NH_3$)) will increase substantially, suggesting an enhanced formation of ammonium aerosols.
Under the moderate control policy (SSP2-45-ECP), the emissions of $SO_2$, $NO_x$, and $NH_3$ in 2050 will
be reduced by 62.7%, 49.0% and 25.0%, respectively with corresponding decreases in $SO_4^{2-}$, $TNO_3$ and
$NH_x$. The predicted pH will increase by ~0.13, and the $NH_4^+$ partitioning ratio will decrease by 0.09,
indicating that relatively more ammonium will exist in the gas phase as $NH_3$. The $NO_3^-$ partitioning ratios
are relatively stable, suggesting its general insensitivity in the predicted pH ranges (Nenes et al., 2020a).
Changes in the $SO_4^{2-}$, $TNO_3$ and $NH_x$ will result in ΔpH of +0.18, -0.05 and -0.02 from 2015 to 2050,
respectively (Figure 6h).
With the strict control policy (SSP1-26-BHE), the emissions of $SO_2$, $NO_x$ and $NH_3$ in 2050 will
decrease by 86.9%, 74.9% and 41.7%, respectively, and the concentrations of $SO_4^{2-}$, $TNO_3$ and $NH_x$
decrease substantially. The pH value will increase continuously by ~0.19 (from 3.36 in 2015 to 3.55 in
2050). Changes in $SO_4^{2-}$ are more important determinants of ΔpH, resulting in ΔpH of +0.28 from 2015
to 2050. Changes in the $TNO_3$ and $NH_x$ are associated with 0.04 and 0.09 decreases in ΔpH, respectively.
Moreover, the $NO_3^-$ and $NH_4^+$ partitioning ratios will decrease by 0.04 and 0.12, respectively, indicating
a benefit of $NH_3$ and $NO_x$ emission controls in mitigating haze pollution in eastern China.
We also note that above analysis based on the historical average Δ aerosol / Δ (precursor emissions) is
subject to uncertainties associated with changes in the atmospheric oxidation capacity, meteorological
conditions, etc. It is only a first-order estimation, and a full examination with 3-D chemical transport
models is recommended in the future.

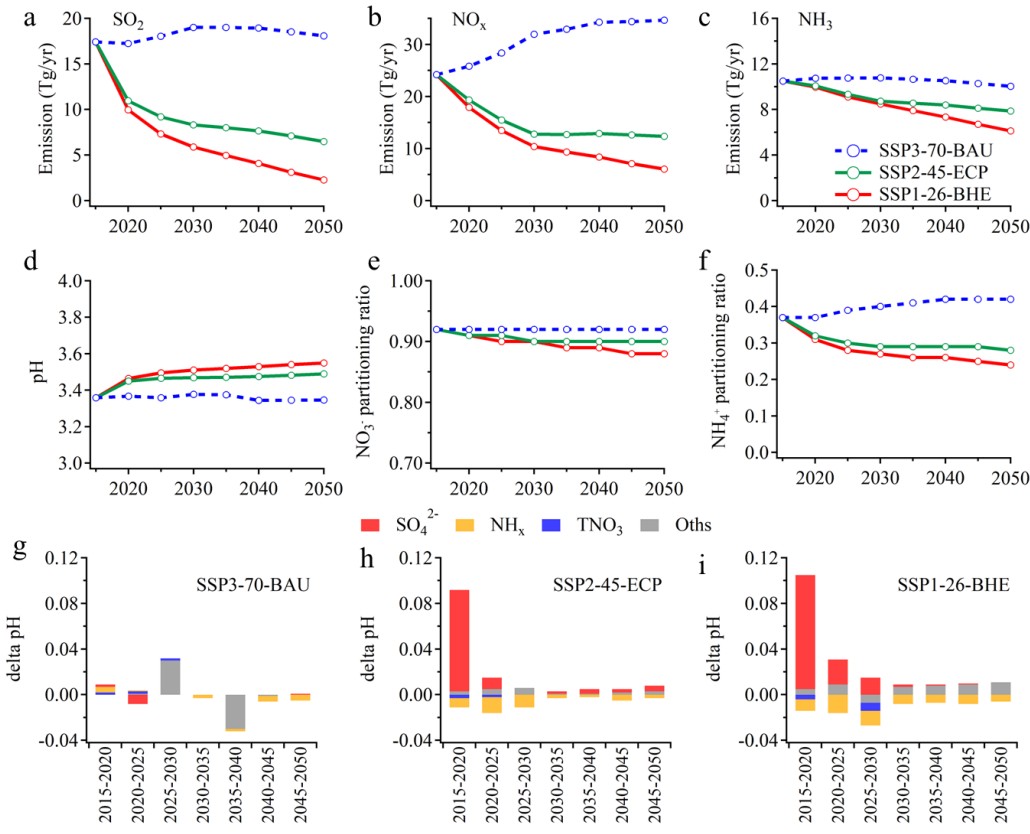

Figure 6. Emissions of $SO_2$ (a), $NO_x$ (b), $NH_3$ (c), predicted pH (d), $NO_3^-$ partitioning ($NO_3^-$ / ($NO_3^-$ + $HNO_3$)) (e) and $NH_4^+$ partitioning ($NH_4^+$ / ($NH_4^+$ + $NH_3$)) (f) in China from 2015 to 2050 under the three scenarios published in Tong et al.(Tong et al., 2020). Predicted contributions of individual factors to the ΔpH under the three scenarios, including SSP3-70-BAU (g), SSP2-45-ECP (h) and SSP1-26-BHE (i). The stacked color bars below the dashed line represent the factors that had negative impacts on ΔpH and the stacked color bars above the dashed line represent the increase in ΔpH. The meanings of the abbreviations: $NH_x$, total ammonia; $TNO_3$, total nitrate; Oths, others.

## 4    Conclusion

The aerosol pH values at an urban site in Shanghai during 2011–2019 were modelled and reported for the first time based on observed gas and aerosol compositions. Although significant variations of aerosol compositions were observed from 2011 to 2019 in the YRD region, the estimated aerosol pH declined only slightly by 0.24. We quantified the contributions from individual factors to the variation of aerosol pH from 2011 to 2019. We found that besides the multiphase buffer effect, $SO_4^{2-}$ and NVCs changes are key in regulating the aerosol pH from 2011 to 2019 in Shanghai. $SO_4^{2-}$ and NVCs showed an overall

opposite effect on aerosol pH, with a contribution of +0.38 and −0.35, respectively.
Distinct seasonal variations in the aerosol pH were observed, with maximum and minimum aerosol
pH of $3.59 \pm 0.57$ in winter and $2.89 \pm 0.49$ in summer, respectively. Seasonal variations in aerosol pH
were mainly driven by the temperature, with the maximum ΔpH of 0.63 between fall and winter. The
diurnal cycle of aerosol pH was driven by the combined effects of temperature and RH which could result
in ΔpH of -0.22 and +0.10, respectively. These results emphasized the importance of meteorological
conditions in controlling the seasonal and diurnal variations of aerosol pH.
To explore the effects of China's future anthropogenic emission control pathways on aerosol pH and
compositions, we chose three different emission reduction scenarios proposed by Tong et al. (2020) for
future haze mitigation, namely SSP3-70-BAU, SSP2-45-ECP and SSP1-26-BHE, as case studies. We
found that under the weak control policy (SSP3-70-BAU), the future aerosol pH and $NO_3^-$ partitioning
ratio will only have subtle changes. While our results show that future aerosol pH will increase under
both strict control policy (SSP1-26-BHE) and moderate control policy (SSP2-45-ECP), the former will
result in a more dramatic increase. The significant increase in aerosol pH is mainly associated with the
decrease in $SO_4^{2-}$. In addition, the increase in aerosol pH with strict control policy and moderate control
policy will lead to relatively more nitrate and ammonium partitioning in the gas phase, which is beneficial
for future $PM_{2.5}$ pollution control. These results highlight the potential effects of precursor reductions on
aerosol pH with future pollution control policy.
**Author Contributions**
HS, HW, and CH conceived and led the study. MZ conducted the field measurements and carried out the
data analysis. MZ and GZ performed model simulations. MZ, HS, HW, CH, GZ, LQ, SZ, DH, YC, JA
discussed the results. LQ, SZ, DH, SL, ST, QW, RY, YM, CC conducted the measurements at the station.
MZ, HS and GZ wrote the manuscript with input from all co-authors.
**Supplement**
The supplement is available in a separate file.
**Competing interests**

The authors declare that they have no conflict of interest.

**Data availability**

The data presented in this paper are available upon request from Hang Su (h.su@mpic.de) and Cheng Huang (huangc@saes.sh.cn).

**Acknowledgement**

This study was supported by the Science and Technology Commission of Shanghai Municipality Fund Project (20dz1204000), the National Key Research and Development Program of China (2018YFC0213800), , the General Fund of National Natural Science Foundation of China (21806108), the National Natural Science Foundation of China (42061134008), the Shanghai Rising-Star Program (19QB1402900) and Shanghai Municipal Bureau of Ecology and Environment Fund Project (2020-03).

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
