# Peer review of "Long-term trends and drivers of aerosol pH in eastern China"

_Atmospheric Chemistry and Physics, 2021_

## Author Comment (AC1)

**Manuscript ID: acp-2021-455**
**TITLE: Long-term trends and drivers of aerosol pH in eastern China**

We thank the editor and the reviewers for the comments concerning our manuscript. They are valuable in helping us improve our manuscript. Below please find our point-by-pint responses to reviewers' comments.

**Comments to the Author**

Major Comments

1. The estimates ALWCo seem unreasonably small (lines 120 - 127)? How was organic aerosol measured? Was it PM$_{2.5}$ as well, or was it PM$_1$?

***Response:*** The concentration of organic aerosol was estimated by multiplying the measured concentration of organic carbon by a factor of 1.6 (Turpin and Lim, 2001). A Thermal/Optical Carbon Aerosol Analyzer (model RT-4, Sunset laboratory Inc.) equipped with a PM$_{2.5}$ cyclone was used for the organic carbon measurement. The annual concentrations of organic carbon in Shanghai were 5.6–10.6μg/m$^3$ from 2011 to 2019, and the relative humidity were 69-75%. $ALWC_o$ was calculated by the following equation (Guo et al., 2015).

$$ALWC_o = \frac{m_{org}\rho_w}{\rho_{org}} \frac{k_{org}}{(\frac{1}{RH}-1)} \qquad (1)$$

where $m_{org}$ is the mass concentration of organic aerosol, $\rho_w$ is the density of water ($\rho_w$=1.0g/cm$^3$), $\rho_{org}$ is the mean density of organics assumed to be 1.4g/cm$^3$)(Guo et al., 2015), and $k_{org}$ is the hygroscopicity parameter of organic aerosol ($k_{org} = 0.087$)(Li et al., 2016). Adopting these values, we estimate that the annual $ALWC_o$ and $ALWC_i$ from 2011 to 2019 are approximately 1.4–2.5μg/m$^3$ and 25.8–35.8μg/m$^3$, respectively. That is $ALWC_o$ accounted for 4.3%–7.5% of the total aerosol liquid water content from 2011 to 2019.

2. I found the convention used in Figures 1b, 3, and 5 very confusing. The pie charts below each figure are useful and seem straightforward to interpret, but the bar charts need substantial editing.   For example, in Figure 1b, the effect of NVCs on the pH trends changes signs with time.   Ultimately, using Fig. S6, I was able to deduce that the positive value associated with NVCs for 2011-2013 meant that NVCs had gone up, and the negative value associated with NVCs for 2013 – 2015 meant that NVCs had gone down.   However, it took far too much time to interpret and is still not easily understandable even after spending much time on it.   The convention used by Tao and Murphy (2021) is much clearer – I suggest edits to follow their approach.

***Response:*** Thanks for the comment. To study the driving factors of aerosol pH, different sensitivity analysis methods have been used in previous studies (Ding et al., 2019; Tao and Murphy, 2021; Zheng et al., 2020). The convention used in Tao and Murphy (2021) defined the base scenario as the average condition, aiming at illustrating the contribution of different factors to the deviation from the base scenario. However, the base scenario can change with the analysis time periods. In comparison, our bar plot here aimed at showing the factor contribution of the ΔpH between two adjacent scenarios (i.e., two continuous years or two continuous hour periods), and is not subject to change in the average conditions. That is, our plots emphasized differently with that used in Tao and Murphy (2021). We've clarified this in the revised figure captions. In addition, to provide more viewpoints, we've added the figures with Tao and Murphy's approach in the supplement following the reviewer's suggestion.

*Changes in manuscript:*

(1) Line 204-206: "Figure 1b shows the contributions of individual factors to the ΔpH from 2011 to 2019. Here the bar plots indicate the factors contributing to the ΔpH between two adjacent scenarios, e.g., 2011 to 2013. See Fig. S9a for the factor contribution to the variation from average conditions."

(2) Line 241-243: "Figure 3 shows the contributions of individual factors to the ΔpH across the four seasons. Here the bar plots indicate the factors contributing to the ΔpH between two adjacent seasons, e.g., spring (MAM) to summer (JJA). See Fig. S9b for the factor contribution to the variation from average conditions."

(3) Line 273-275: "Figure 5 shows the effects of individual factors to the ΔpH between day and night. Here the bar plots indicate the factors contributing to the ΔpH between two adjacent hour periods, e.g., 0:00 to 6:00. See Fig. S9c for the factor contribution to the variation from average conditions."

We've revised Fig. 1b, 3 and 5 in the manuscript and added more description in the captions. For example, we've revised Fig. 3 into:

[Figure]

**Figure R1** *(revised Fig. 3 in the manuscript)*. **Contributions of individual factors to the ΔpH across the four seasons.** Here the bar plots indicate the factors contributing to the ΔpH between two adjacent seasons, e.g., spring (MAM) to summer (JJA). The meanings of the abbreviations: RH, relative humidity; Temp, temperature; NVCs, non-volatile cations; $NH_x$, total ammonia; $TNO_3$, total nitrate.

*Changes in* **supplement of** *manuscript:*

Line 111-117: We've added the Figures S9 in the revised supplement as:

[Figure]

**Figure R2** *(added as Fig. S9 in the revised supplement)*. **Fractional contribution of individual factors to the variations in aerosol pH from average conditions (i.e., averages of all observational data) during 2011–2019.** (a) Annual variation; (b) Seasonal variation, and (c) diurnal variation. The meanings of the abbreviations: RH, relative humidity; Temp, temperature; NVCs, non-volatile cations; $NH_x$, total ammonia; $TNO_3$, total nitrate.

3. Discussion about the limited effects of future emissions control measures on haze pollution (e.g., line 35-36, 298-299) is just wrong. Although the partitioning of $NH_3$ and $HNO_3$ may shift towards the particulate phase in the future, it does not mean their total PM concentration has increased. If the total concentration (i.e., $NH_3 + NH_4^+$) decreased enough, then a shift in partitioning towards the particle phase could still occur with a decrease in the aerosol $NH_4^+$. This discussion would be much better with associated predictions of the $PM_{2.5}$, $NH_4^+$, $SO_4^{2-}$, and NO3- aerosol concentrations.

*Response:* Thanks for the comment. We agree that the precursor decrease will finally lead to a PM decrease. Here we are discussing about the efficiency of PM reduction concentrations against the precursor reduction concentrations. To further clarify our points, we've revised the corresponding manuscript and figures with more detailed explanations. In addition, we've added the prediction of the changes in major chemical components ($NH_4^+$, $SO_4^{2-}$, $NO_3^-$ and $Cl^-$) as Fig. 6g-i following the reviewer's suggestion. See detailed modifications below.

***Changes in manuscript:***

**(1)** Line 34-38: We've revised the statement into: "The corresponding aerosol pH in eastern China is estimated to increase by ~0.9, and the reduction in particle phase $NO_3^-$ and $NH_4^+$ is less than the reduced amount of total $HNO_3$ and total $NH_3$. This suggests a reduced benefit of $NH_3$ and $NO_x$ emission control in mitigating haze pollution in eastern China."

**(2)** Discussions in section 3.4: We've revised Fig. 6 and the corresponding discussions into (Line 320-366 in the revised section 3.4):

"Under the reference scenario of SSP3-70-BAU with weak control policy (blue lines in Fig. 6 a-f), $SO_2$ and NOx are predicted to increase, while the NHx is relatively stable. Correspondingly, both $SO_4^{2-}$ and $NO_3^-$ will increase, and $NH_4^+$ will also increase in response (Fig. 6g). Considering the stable NHx, $NH_4^+$ partition ratio ($NH_4^+ / (NH_4^+ + NH_3)$) will increase. In comparison, there is little change in aerosol pH and the predicted $NO_3^-$ partition ratio ($NO_3^- / (NO_3^- + HNO_3)$).

Under the moderate control policy (SSP2-45-ECP), the emissions of $SO_2$, $NO_x$, and $NH_3$ in 2050 will be reduced by 62.7%, 49.0% and 25.0%, respectively. Correspondingly, $SO_4^{2-}$, $NO_3^-$ and $NH_4^+$ will all decrease (Fig. 6h), with a total PM reduction of ~14.4 $\mu g\ m^{-3}$. Moreover, the predicted pH will increase by ~0.5, and the $NO_3^-$ and $NH_4^+$ partition ratios will decrease by 0.14 and 0.23, respectively (green lines in Fig. 6d-f). That is, more nitrate and ammonium will exist in the gas phase as $HNO_3$ and $NH_3$, thus the reduced $NH_4^+$ and $NO_3^-$ is higher than the reduced NHx and TNO_3, which is a control bonus in terms of reduced PM per reduced emissions for this scenario.

With the strict control policy (SSP1-26-BHE), the emissions of $SO_2$, $NO_x$ and $NH_3$ in 2050 will decrease by 86.9%, 74.9% and 41.7%, respectively. Its effect on PM reductions resembles that of the moderate one (SSP2-45-ECP) before 2040. Afterwards, however, the $NO_3^-$ partition ratio increased despite the increasing pH, and reached near 1 in 2050 (Fig. 6 d, e). On second check, we found this pattern is due to the sharp decrease in $SO_4^{2-}$ and constant NVCs. After 2040, there will be a major anion deficit considering the non-volatile species only (sulfate and $Ca^{2+}$, $K^+$, $Mg^{2+}$), and therefore more $NO_3^-$ will be captured by the NVCs to the particle phase. As a result, $NO_3^-$ partition ratio even increased from 0.92 in 2015 to 1.00 in 2050. Although $NH_4^+$ partition ratio showed a continuous decrease, in 2050 both the reduced $NH_4^+$ and $NO_3^-$ is smaller than the reduced NHx and TNO_3 (Fig. 6i). That is in contrast with the effect of the moderate one (SSP2-45-ECP). Correspondingly, the total reduced PM is only slightly larger for the strict SSP1-26-BHE policy (~18.6 $\mu g\ m^{-3}$) than the moderate SSP2-45-ECP policy (~14.4 $\mu g\ m^{-3}$) indicating a reduced efficiency in terms of PM controls in responses to the emission controls. This would suggest a reduced benefit of $NH_3$ and $NO_x$ emission control in mitigating haze pollution in eastern China, especially after 2040."

[Figure]

**Figure R3** *(revised Fig. 6 in the manuscript)*. Emissions of $SO_2$ (a), $NO_x$ (b), $NH_3$ (c), predicted pH (d), $NO_3^-$ partition ($NO_3^- / (NO_3^- + HNO_3)$) (e) and $NH_4^+$ partition ($NH_4^+ / (NH_4^+ + NH_3)$) (f) in China from 2015 to 2050 under the three scenarios published in Tong et al.(2020). Predicted the changes in major chemical components ($NH_4^+$, $SO_4^{2-}$, $NO_3^-$ and $Cl^-$) and reductions in $TNO_3$ and $NHx$ under the three scenarios, including SSP3-70-BAU (g), SSP2-45-ECP (h) and SSP1-26-BHE (i).

4. The Conclusions section needs substantial revision. A brief recap is ok, but Section 4 is mostly redundant with the prior section. Rather than just reiterating what has already been said, more discussion of the significance of the work is warranted.

***Response:*** Thanks for the comment. We rewrote the conclusions section. Please see the following changes.

***Changes in manuscript:***

Line 369-415: The specific modifications are as follows:

"The aerosol pH values at an urban site in Shanghai during 2011–2019 were calculated using ISORROPIA II. The trend analysis of aerosol pH in Shanghai during 2011–2019 was reported firstly based on observed gas and aerosol composition. Although significant variations of aerosol compositions were observed from 2011 to 2019 in YRD region, the aerosol pH estimated by model only slightly declined by 0.24 unit. We quantified the contributions from individual factors on the variation of aerosol pH from 2011 to 2019. We revealed that besides the multiphase buffer effect, the opposite effects of $SO_4^{2-}$ and non-volatile cations changes with a contribution of +0.38 and −0.35 unit on aerosol pH, respectively play a key role in determining the moderate pH trend from 2011 to 2019.

Distinct seasonal variations in the aerosol pH were observed, with maximum and minimum aerosol pH of 3.59 ± 0.57 in winter and 2.89 ± 0.49 in summer, respectively. Seasonal variations in aerosol pH

were mainly driven by the temperature, with the max ΔpH of 0.63 existed between fall and winter. The diurnal cycle of particle pH was driven by the combined effects of temperature and relative humidity which could result in ΔpH of -0.22 and +0.10 units, respectively. These results emphasized the importance of meteorological conditions in controlling the seasonal and diurnal variations of aerosol pH.

Finally, to explore the effects of China's future anthropogenic emission control pathways on aerosol pH and compositions, we chose three different emission reduction scenarios proposed by Tong et al.(2020) for future haze mitigation, naming SSP3-70-BAU, SSP2-45-ECP and SSP1-26-BHE as case studies. We estimated that the future trend of aerosol pH and $NO_3^-$ partition ratio will change little under the weak control policy (SSP3-70-BAU), while $SO_4^{2-}$, $NO_3^-$ and $NH_4^+$ will increase substantially. The results also demonstrate that future aerosol pH will increase under both strict control policy (SSP1-26-BHE) and moderate control policy (SSP2-45-ECP), but more drastically under former scenario. The significant increase in aerosol pH with the strict control policy will lead to the reduced aerosol $NH_4^+$ and $NO_3^-$ is smaller than the reduced amount of total NH3 and total HNO3, which is in contrast with effect of the moderate control policy. This suggests that a reduced efficiency in terms of PM controls in responses to the emission controls with the strict control policy. These results highlight the importance of proportional reductions in precursors and follow-up variations in aerosol pH in future pollution control policy."

5. Finally, the entire manuscript needs to be edited for language consistency – specifically, verb tenses change within and between paragraphs. There are too many instances to list here.

*Response:* Thanks for the comment. The language consistency in the manuscript has been polished, please see the modifications in the revised manuscript.

Technical/Minor Comments
1. Line 25: define all acronyms on their first use (e.g., NVCs, YRD)
2. Line 28-29: sentence needs grammatical editing.
3. Line 39: suggest deleting "studies"
4. Line 44-45: cite also Tilgner et al. (2021) in this group.
5. Line 73: cite also Vasilakos et al. (2018) and Nenes et al. (2020)
6. Line 77: "composition" should be singular
7. Line 78: suggest changing "characterizing" to "characterize"
8. Line 96: change "to be" to "were"
9. Line 96: "calibration" is not the right term here – LiBr is used as an internal standard
10. Line 102: due to unmeasured species (organic acid ions, carbonate) – it is quite possible to measure the aerosol inorganic composition accurately and not achieve an ion balance. Given what we know about organic acid concentrations, it is actually surprising that such a balance is observed.
11. Line 104: cite also Stieger et al. (2018)
12. Line 106: suggest deleting "techniques"
13. Line 107: give the instrument(s) used to measure T and RH
14. Line 128: cite also Battaglia Jr., et al. (2019)
15. Line 237-238: the diurnal behavior of aerosol pH is not just consistent with Beijing, but is far more consistent (qualitatively) with many other locations like the SE USA (Guo et al., 2015), eastern US (Battaglia et al., 2017), Chicago (Battaglia et al., 2017), which shows the important influences of T and RH on aerosol pH.
16. Line 274: "active actions" should be changed

**Response:** Thanks for the comments. We've revised the manuscript based on the above comments. Please see the following changes.

***Changes in manuscript:***

(1) Lines 24-26: We added the definition of NVCs and YRD. Please see the details as follows: "The implementation of the Air Pollution Prevention and Control Action Plan led to -35.8%, -37.6%, -9.6%, -81.0% and 1.2% changes of $PM_{2.5}$, $SO_4^{2-}$, $NH_x$, non-volatile cations (NVCs) and $NO_3^-$ in Yangtze River Delta (YRD) region during this period."

(2) Lines 26-30: We rewrote this sentence, please see the details as follows: "Different from the fast changes of aerosol compositions due to the implementation of the Air Pollution Prevention and Control Action Plan, aerosol pH showed a moderate change of -0.24 unit over the 9 years. Besides the multiphase buffer effect, the opposite effects from the changes of $SO_4^{2-}$ and non-volatile cations played key roles in determining the moderate pH trend, contributing to a change of +0.38 and −0.35 unit, respectively"

(3) Line 41: We deleted the word "studies", please see the details as follows: "Aerosol acidity is an important parameter in atmospheric chemistry."

(4) Line 44-47: We added the cite of Tilgner et al. (2021) as: "Aerosol acidity has attracted an increasing concern in recent years because of its impacts on the thermodynamics of gas-particle partitioning, pH-dependent condensed-phase reactions and trace metal solubility(Cheng et al., 2016; Fang et al., 2017; Guo et al., 2017; Guo et al., 2016; He et al., 2018; Song et al., 2018; Su et al., 2020; Tilgner et al., 2021; Weber et al., 2016)."

(5) Line 73-76: We added the cites of Vasilakos et al. (2018) and Nenes et al. (2020), please see the details as follows: "Aerosol pH may change due to the significant changes of the chemical composition in $PM_{2.5}$, which may feedback to the multiphase formation pathways of aerosols such as sulfate, nitrate and ammonium (Cheng et al., 2016; Nenes et al., 2020; Vasilakos et al., 2018)"

(6) Line 78-80: We edited this sentence as: "A thermodynamic model, ISORROPIA II (version 2.1) (Fountoukis and Nenes, 2007) was applied to estimate the pH based on 9-year continuous online measurements of $PM_{2.5}$ composition at an urban site in Shanghai."

(7) Line 80-81: We edited this sentence, please see the details as follows: "The main purposes of this study are to: (1) characterize the long-term trend of aerosol pH;"

(8) Line 98-99: We edited this sentence into: "To better track the retention time changes of different ion species and ensure their concentrations were measured successfully,"

(9) Line 98-100: We rewrote this sentence into: "To better track the retention time changes of different ion species and ensure their concentrations were measured successfully, an internal standard check was conducted every hour with Lithium Bromide (LiBr) standard solution (Qiao et al., 2014; Zhou et al., 2016)."

(10) Line 105-108: Indeed, the measurements of organic acid ions were lacked in our study. However, we find that in previous studies, the concentrations of organic acid in Shanghai area were low. Ding et al., (2021) found that total dicarboxylic acids in Chongming Island in Shanghai during the day and night was $375\pm282$ ng/m$^3$ and $341\pm270$ ng/m$^3$, respectively, and the ketocarboxylic acids ranged from 3.3 ng/m$^3$ to 125 ng/m$^3$. Yao et al., (2002) also found that the sum of oxalate, malonate and succinate only account for 0.3-2% of the total mass of the water-soluble ions in Shanghai. The concentrations of organic ions were significantly lower than that of $SO_4^{2-}$ and $NO_3^-$, which were the main anions in aerosol of Shanghai. Due to the low concentrations of organic acids in $PM_{2.5}$, they

may have minor effects on ion balance. Meanwhile, we also find that the average equivalent ratios of cation/anion(C/A) were close to unity in many cities of China (Huang et al., 2014; Shen et al., 2010; Sun et al., 2006; Zhang et al., 2018). In our study, the modelled and measured $NH_3$ and $NH_4^+$ concentrations were in good agreement based on observed aerosol composition, further indicating that the measurement of the ions was accurate. We rewrote this sentence as: "The correlation between cation and anion was strong ($R^2$=0.94), with a slope of 1.00, indicating that these ion species were charge balanced and well represented major components in $PM_{2.5}$."

(11) Line 108-110: We added the cites of Stieger et al. (2018) as: "In previous studies, intercomparison experiments between MARGA and filter-based method have been carried out, and the data measured by MARGA showed acceptable accuracy and precision (Rumsey et al., 2014; Huang et al., 2014; Stieger et al., 2018)"

(12) Line 110-113: We edited this sentence as: "The mass concentrations of $PM_{2.5}$ were simultaneously measured using an on-line beta attenuation PM monitor (FH 62 C14 series, Thermo Fisher Scientific) at a time resolution of 5 min."

(13) Line 112-114: We added the instrument information used to measure T and RH, please see the details as follows: "The temperature and RH were also measured using meteorological parameters monitor (Metone 597, Met One Instruments) at a time resolution of 1 min."

(14) Line 135-137: We added the cite of Battaglia Jr., et al. (2019) in this sentence as: "The use of $ALWC_i$ to predict pH is therefore fairly accurate and common(Battaglia Jr et al., 2019; Battaglia et al., 2017; Ding et al., 2019)"

(15) Line 270-272: We rewrote this sentence into: "Figure 4 shows the diurnal variations in the aerosol pH and its potential drivers. Aerosol pH in Shanghai exhibits notable diurnal variations, being higher during nighttime.

(16) Line 314-315: We rewrote this sentence into: "SSP3-70-BAU is a reference scenario that without additional efforts to constrain emissions."

(17) Line 330-333: We edited this sentence into: "Moreover, the predicted pH will increase by ~0.5, and the $NO_3^-$ and $NH_4^+$ partition ratios will decrease by 0.14 and 0.23, respectively (green lines in Fig. 6d-f)."

**Reference**

Battaglia Jr, M. A., et al., 2019. Effects of water-soluble organic carbon on aerosol pH. Atmospheric Chemistry and Physics. 19, 14607-14620.

Battaglia, M. A., et al., 2017. Effect of the Urban Heat Island on Aerosol pH. Environmental Science & Technology. 51, 13095-13103.

Cheng, Y., et al., 2016. Reactive nitrogen chemistry in aerosol water as a source of sulfate during haze events in China. Science Advance.

Ding, J., et al., 2019. Aerosol pH and its driving factors in Beijing. Atmospheric Chemistry and Physics. 19, 7939-7954.

Ding, Z., et al., 2021. Summertime atmospheric dicarboxylic acids and related SOA in the background region of Yangtze River Delta, China: Implications for heterogeneous reaction of oxalic acid with sea salts. Sci Total Environ. 757, 143741.

Fang, T., et al., 2017. Highly Acidic Ambient Particles, Soluble Metals, and Oxidative Potential: A Link between Sulfate and Aerosol Toxicity. Environ Sci Technol. 51, 2611-2620.

Fountoukis, C., Nenes, A., 2007. ISORROPIA II: a computationally efficient thermodynamic equilibrium model for K+–Ca2+–Mg2+–NH+

4 –Na+–SO2− 4 –NO− 3 –Cl−–H2O aerosols. Atmospheric Chemistry and Physics. 7, 4639-4659.

Guo, H., et al., 2017. Fine particle pH and gas–particle phase partitioning of inorganic species in Pasadena, California, during the 2010 CalNex campaign. Atmospheric Chemistry and Physics. 17, 5703-5719.

Guo, H., et al., 2016. Fine particle pH and the partitioning of nitric acid during winter in the northeastern United States. Journal of Geophysical Research: Atmospheres. 121, 10,355-10,376.

Guo, H., et al., 2015. Fine-particle water and pH in the southeastern United States. Atmospheric Chemistry and Physics. 15, 5211-5228.

He, P., et al., 2018. Isotopic constraints on heterogeneous sulfate production in Beijing haze. Atmospheric Chemistry and Physics. 18, 5515-5528.

Huang, X. H. H., et al., 2014. Characterization of PM2.5 Major Components and Source Investigation in Suburban Hong Kong: A One Year Monitoring Study. Aerosol and Air Quality Research. 14, 237-250.

Li, C., et al., 2016. Physiochemical properties of carbonaceous aerosol from agricultural residue burning: Density, volatility, and hygroscopicity. Atmospheric Environment. 140, 94-105.

Nenes, A., et al., 2020. Aerosol pH and liquid water content determine when particulate matter is sensitive to ammonia and nitrate availability. Atmospheric Chemistry and Physics. 20, 3249-3258.

Qiao, L., et al., 2014. PM2.5 constituents and hospital emergency-room visits in Shanghai, China. Environ Sci Technol. 48, 10406-14.

Shen, Z., et al., 2010. Chemical Characteristics of Fine Particles (PM1) from Xi'an, China. Aerosol Science and Technology. 44, 461-472.

Song, S., et al., 2018. Fine-particle pH for Beijing winter haze as inferred from different thermodynamic equilibrium models. Atmospheric Chemistry and Physics. 18, 7423-7438.

Su, H., et al., 2020. New Multiphase Chemical Processes Influencing Atmospheric Aerosols, Air Quality, and Climate in the Anthropocene. Acc Chem Res. 53, 2034-2043.

Sun, Y., et al., 2006. Chemical Characteristics of PM2.5 and PM10 in Haze-Fog Episodes in Beijing. Environmental Science & Technology 40, 3128-3155.

Tao, Y., Murphy, J. G., 2021. Simple Framework to Quantify the Contributions from Different Factors Influencing Aerosol pH Based on NHx Phase-Partitioning Equilibrium. Environ Sci Technol.

55, 10310-10319.

Tilgner, A., et al., 2021. Acidity and the multiphase chemistry of atmospheric aqueous particles and clouds. Atmospheric Chemistry and Physics. 21, 13483-13536.

Tong, D., et al., 2020. Dynamic projection of anthropogenic emissions in China: methodology and 2015–2050 emission pathways under a range of socio-economic, climate policy, and pollution control scenarios. Atmospheric Chemistry and Physics. 20, 5729-5757.

Turpin, B. J., Lim, H.-J., 2001. Species Contributions to PM2.5 Mass Concentrations: Revisiting Common Assumptions for Estimating Organic Mass. Aerosol Science and Technology. 35, 602-610.

Vasilakos, P., et al., 2018. Understanding nitrate formation in a world with less sulfate. Atmospheric Chemistry and Physics. 18, 12765-12775.

Weber, R. J., et al., 2016. High aerosol acidity despite declining atmospheric sulfate concentrations over the past 15 years. Nature Geoscience. 9, 282-285.

Yao, X., et al., 2002. The water-soluble ionic composition of PM2.5 in Shanghai and Beijing, China. Atmospheric Environment. 36, 4223-4234.

Zhang, J., et al., 2018. Seasonal variation and size distributions of water-soluble inorganic ions and carbonaceous aerosols at a coastal site in Ningbo, China. Sci Total Environ. 639, 793-803.

Zheng, G., et al., 2020. Multiphase buffer theory explains contrasts in atmospheric aerosol acidity. Science 369, 1374-1377.

Zhou, M., et al., 2016. Chemical characteristics of fine particles and their impact on visibility impairment in Shanghai based on a 1-year period observation. J Environ Sci (China). 48, 151-160.

---

## Author Comment (AC2)

**Manuscript ID: acp-2021-455**
**TITLE: Long-term trends and drivers of aerosol pH in eastern China**

We thank the editor and the reviewers for the comments concerning our manuscript. They are valuable in helping us improve our manuscript. Below please find our point-by-pint responses to reviewers' comments.

**Comments to the Author**

1. Overall, my main concern with the manuscript is that the methods section is much too brief. The authors need to provide a lot more explanation of how they generated the data that are presented in Figures 1, 3, 5, and 6. Since each of there figures represents perturbations or trends to some previous averaging period, the way in which the data are averaged (and perturbed) needs to be explained more clearly. For the long-term trend in Figure 1, the approach seems fairly obvious, but the way in which the seasonal and diel cycles are formulated in Figure 3 and 5 is quite confusing.

*Response:* Thanks for the comment. We've added the more description of methods in the revised manuscript. Please see the following changes.

*Changes in manuscript:*

(1) Line 150-178: We rewrote the section 2.3 as:

"**2.3 Drivers of aerosol pH variations**
To investigate the factors that drive changes in aerosol pH, sensitivity tests of pH variations to different factors, including temperature, RH, $SO_4^{2-}$, $TNO_3$, $NH_x$, $Cl^-$ and NVCs, were performed with the one-at-a-time method. For illustration, assume the aerosol pH estimated under scenario I ($pH_I$) differs with that under scenario II ($pH_{II}$), and the pH difference, $\Delta pH = pH_{II} - pH_I$, are caused by the variations in the factors listed above. To quantify the contributions of individual factors, we varied the factor $i$ from the level in scenario I to that in scenario II while keeping the other factors fixed. The corresponding pH changes, $\Delta pH_i$, are assumed to represent the contribution of this individual factor change to the overall aerosol pH variations. The unresolved contributors to pH differences, i.e., $\Delta pH - \sum_i \Delta pH_i$, are attributed to "others", which may represent the contribution of covariations between the factors. This method is applied in Fig. 1b, Fig. 3 and Fig. 5, where the corresponding scenarios represent the average conditions in different years (Fig. 1b), seasons (Fig. 3) or diurnal periods (Fig. 5)."

(2) Line 204-206: We added some descriptions as: "Figure 1b shows the contributions of individual factors to the $\Delta pH$ from 2011 to 2019. Here the bar plots indicate the factors contributing to the $\Delta pH$ between two adjacent scenarios, e.g., 2011 to 2013. See Fig. S9a for the factor contribution to the variation from average conditions."

(3) Line 241-245: We added some descriptions into: "Figure 3 shows the contributions of individual factors to the $\Delta pH$ across the four seasons. Here the bar plots indicate the factors contributing to the $\Delta pH$ between two adjacent seasons, e.g., spring (MAM) to summer (JJA). See Fig. S9b for the factor contribution to the variation from average conditions. The aerosol pH was calculated from the mean averages of input parameters in four seasons, and the $\Delta pH$ was estimated by varying one factor while holding the other factors fixed in different seasons."

(4) Line 273-278: We added some descriptions into "Figure 5 shows the effects of individual factors to

the ∆pH between day and night. Here the bar plots indicate the factors contributing to the ∆pH between two adjacent hour periods, e.g., 0:00 to 6:00. See Fig. S9c for the factor contribution to the variation from average conditions. The aerosol pH was calculated from the mean averages of input parameters in 0:00, 6:00, 12:00 and 18:00, and ∆pH was estimated by varying one factor while holding the other factors fixed in different hours."

2. P1 L33-36 It is hard to understand the meaning of this sentence. Are the authors suggesting that $NH_3$ and $NO_x$ emission controls are not going to be effective in the time period leading up to 2050, or that they won't be effective after that point? Further – is it appropriate to examine the values just on their own – what would happen if only $SO_2$ reductions were implemented? Presumably $NH_4^+$ and $NO_3^-$ would increase much more.

*Response:* Thanks for the comment. Here we are discussing about the efficiency of PM reduction concentrations against the precursor reduction concentrations. To further clarify our points, we've revised the corresponding manuscript, and added the prediction of the changes in major chemical components ($NH_4^+$, $SO_4^{2-}$, $NO_3^-$ and $Cl^-$) as Fig. 6g-i. See detailed modifications below.

*Changes in manuscript:*

(1) Line 34-38: We've revised the statement into: "The corresponding aerosol pH in eastern China is estimated to increase by ~0.9, and the reduction in particle phase $NO_3^-$ and $NH_4^+$ is less than the reduced amount of total $HNO_3$ and total $NH_3$. This suggests a reduced benefit of $NH_3$ and $NO_x$ emission control in mitigating haze pollution in eastern China."

(2) Discussions in section 3.4: We've revised Fig. 6 and the corresponding discussions into (Line 320-366 in the revised section 3.4):

"Under the reference scenario of SSP3-70-BAU with weak control policy (blue lines in Fig. 6 a-f), $SO_2$ and NOx are predicted to increase, while the NHx is relatively stable. Correspondingly, both $SO_4^{2-}$ and $NO_3^-$ will increase, and $NH_4^+$ will also increase in response (Fig. 6g). Considering the stable NHx, $NH_4^+$ partition ratio ($NH_4^+ / (NH_4^+ + NH_3)$) will increase. In comparison, there is little change in aerosol pH and the predicted $NO_3^-$ partition ratio ($NO_3^- / (NO_3^- + HNO_3)$).

Under the moderate control policy (SSP2-45-ECP), the emissions of $SO_2$, $NO_x$, and $NH_3$ in 2050 will be reduced by 62.7%, 49.0% and 25.0%, respectively. Correspondingly, $SO_4^{2-}$, $NO_3^-$ and $NH_4^+$ will all decrease (Fig. 6h), with a total PM reduction of ~14.4 μg m$^{-3}$. Moreover, the predicted pH will increase by ~0.5, and the $NO_3^-$ and $NH_4^+$ partition ratios will decrease by 0.14 and 0.23, respectively (green lines in Fig. 6d-f). That is, more nitrate and ammonium will exist in the gas phase as $HNO_3$ and $NH_3$, thus the reduced $NH_4^+$ and $NO_3^-$ is higher than the reduced NHx and TNO₃, which is a control bonus in terms of reduced PM per reduced emissions for this scenario.

With the strict control policy (SSP1-26-BHE), the emissions of $SO_2$, $NO_x$ and $NH_3$ in 2050 will decrease by 86.9%, 74.9% and 41.7%, respectively. Its effect on PM reductions resembles that of the moderate one (SSP2-45-ECP) before 2040. Afterwards, however, the $NO_3^-$ partition ratio increased despite the increasing pH, and reached near 1 in 2050 (Fig. 6 d, e). On second check, we found this pattern is due to the sharp decrease in $SO_4^{2-}$ and constant NVCs. After 2040, there will be a major anion deficit considering the non-volatile species only (sulfate and $Ca^{2+}$, $K^+$, $Mg^{2+}$), and therefore more $NO_3^-$ will be captured by the NVCs to the particle phase. As a result, $NO_3^-$ partition ratio even increased from 0.92 in 2015 to 1.00 in 2050. Although $NH_4^+$ partition ratio showed a

continuous decrease, in 2050 both the reduced $NH_4^+$ and $NO_3^-$ is smaller than the reduced NHx and $TNO_3$ (Fig. 6i). That is in contrast with the effect of the moderate one (SSP2-45-ECP). Correspondingly, the total reduced PM is only slightly larger for the strict SSP1-26-BHE policy (~18.6 μg m$^{-3}$) than the moderate SSP2-45-ECP policy (~14.4 μg m$^{-3}$) indicating a reduced efficiency in terms of PM controls in responses to the emission controls. This would suggest a reduced benefit of $NH_3$ and $NO_x$ emission control in mitigating haze pollution in eastern China, especially after 2040."

[Figure]

**Figure R1** *(revised Fig. 6 in the manuscript)*. **Emissions of $SO_2$ (a), $NO_x$ (b), $NH_3$ (c), predicted pH (d), $NO_3^-$ partition ($NO_3^-$ / ($NO_3^-$ + $HNO_3$)) (e) and $NH_4^+$ partition ($NH_4^+$ / ($NH_4^+$ + $NH_3$)) (f) in China from 2015 to 2050 under the three scenarios published in Tong et al.(2020). Predicted the changes in major chemical components ($NH_4^+$, $SO_4^{2-}$, $NO_3^-$ and $Cl^-$) and reductions in $TNO_3$ and NHx under the three scenarios, including SSP3-70-BAU (g), SSP2-45-ECP (h) and SSP1-26-BHE (i).**

In addition, following the reviewer's suggestion, we also examined the changes of $SO_4^{2-}$, $NH_4^+$ and $NO_3^-$ aerosol concentrations if only one gaseous precursor reduction is implemented (Figure R2). As shown in Figure R2, if only $SO_2$ reduction is implemented, $NH_4^+$ and $SO_4^{2-}$ concentrations will show a significant decrease from 2015 to 2050, while $NO_3^-$ concentration will generally keep constant between 2015-2040 and then increase slightly during 2040-2050. If only NHx reduction is implemented, $NH_4^+$ and $NO_3^-$ concentrations will be slightly reduced while $SO_4^{2-}$ concentration will remain unchanged from 2015 to 2050. However, if only $NO_x$ reduction is implemented, both $NH_4^+$ and $NO_3^-$ concentrations will be significantly reduced.

[Figure]

**Figure R2 The mass concentrations of $SO_4^{2-}$, $NH_4^+$ and $NO_3^-$ aerosol from 2015 to 2050 if only $SO_2$(a), $NH_x$(b) and $NO_x$(c) reductions were implemented under the SSP1-26-BHE scenario published in Tong et al.(Tong et al., 2020)**

3. The data in Figure S2 look much more tightly correlated in the later years. Can the authors comment on whether this reflects improvements in the accuracy/precision of the measurements or whether the relative importance on the measured ions to the overall ion balance may have changed?

***Response:*** Thanks for the comment. Indeed, the occurrence possibility of of high C/A ratios (larger than 1.1) is significantly lower during 2017-2019 than that during 2011-2013 and 2014-2016 (Fig. R3), which is the major reason of the more scattered data in 2011-2016 than that in 2017-2019. To investigate into the potential reasons, we further compared the chemical profiles at different C/A ratio levels (Fig. R4). We found that the high C/A ratio samples are mainly driven by the increased NVCs, while the other chemical compositions (anions and $NH_4^+$) show little dependence with C/A ratios. Accordingly, the decreased fraction of sample with high C/A ratios in recent years is due to the significant decreases in $Ca^{2+}$, $K^+$ and $Mg^{2+}$ from 2011 to 2019, with the annual decrease rates of 14.4%, 30.0% and 15.2%, respectively (Fig. R5). This decrease in NVCs can be attributed to the nationwide control of fugitive dust and biomass burning (An et al., 2021; Cheng et al., 2019; Ding et al., 2019). That is, the less scattered data in 2017-2019 is due to the decreased occurrence of dusty periods in recent years. We've added this information into Supplement as Figure S3-S5 and more descriptions in the section 2.1. Please see the following changes.

***Changes in manuscript:***

Line 102-105: We added some descriptions into: "Figure S2 compares the sum of $SO_4^{2-}$, $NO_3^-$ and $Cl^-$ with the sum of $NH_4^+$, $Na^+$, $K^+$, $Ca^{2+}$ and $Mg^{2+}$ in $neq/m^3$ to check the charge balance. Data in 2011-2016 were more scattered than that in 2017-2019, mainly due to the significant decreases in $Ca^{2+}$, $K^+$ and $Mg^{2+}$ from 2011 to 2019 (Fig S3-S5)."

***Changes in supplement of manuscript:***

Line 71-79: We've added the Figures S3-S5 in the revised supplement as:

[Figure]

**Figure R3 (*added as Fig. S3 in the revised supplement*) The box plots of Cation/Anion ratios during 2011-2013, 2014-2016, and 2017-2019**

[Figure]

**Figure R4 (*added as Fig. S4 in the revised supplement)* Average equivalence concentrations of cation and anion at different level of Cation/Anion ratio**

[Figure]

**Figure R5 (*added as Fig. S5 in the revised supplement*) Monthly mean concentrations of Ca²⁺, K⁺ and Mg²⁺ from 2011 to 2019**

4. The sensitivity tests mentioned in Section 2.3 (lines 143-152) and in Figure S4 are not sufficiently well-described. What did the authors alter and what did they hold constant in each test? The results appear to span a different range of ALWC for each variable. In general, I did not find this section added much to the manuscript, and only made me confused about the method. I suggest removing this section unless it can be much more clearly explained.

*Response:* Thanks for the comment. We've deleted this paragraph following the reviewer's suggestion. We tried to give a general view in how a factor is related to aerosol pH (i.e., positively or negatively) when other factors are kept constant, but we agree with the reviewer in that this part is not closely related to the main points in this study.

5. The language in Section 3.4 is a little confusing because the authors describe the changes in absolute amounts of particle and gas phase ammonium and nitrate somewhat interchangeably with their partitioning ratios and it becomes hard to keep track of what metric is being described. It might be more clear to focus first on the absolute particle phase amounts and then explain the changes in the context of the partitioning. The sharp increase in the particle phase partitioning of nitrate between 2040 and 2050 is quite difficult to understand in the BHE scenario – what explains it?

*Response:* Thanks for the comment. We've added more panels in Fig. 6 and rewrote this section. Please see our response to your comment #2.

6. Figure S6 – The caption should explain why are the rates of change only calculated starting in 2013 when the data record starts in 2011. And why a separate slope is calculated for the latter part of the record.

*Response:* Thanks for the comment. We analyzed the trend before and after 2013 separately as the Air Pollution Prevention and Control Action Plan is implemented in 2013. We've added more explanation in

the caption of Figure S8.

***Changes in supplement of manuscript:***

Line 100-106: We revised the caption of this figure in the supplement of manuscript, the specific modifications are as follows:

[Figure]

**Figure R6 (*revised as Fig. S8 in the revised supplement*) Monthly mean of PM$_{2.5}$, SO$_4^{2-}$, NO$_3^-$, NH$_x$, Cl$^-$ and NVCs from 2011 to 2019.** The years of 2011-2013, 2013-2017 and 2017-2019 represent the Pre-Action Plan, Action Plan and Post-Action Plan period, respectively. Here we focused on the changes in trends between the Action Plan (2013-2017; black dashed lines) and Post-Action Plan (2017-2019; green dashed lines) periods.

**Reference**

An, J., et al., 2021. Emission inventory of air pollutants and chemical speciation for specific anthropogenic sources based on local measurements in the Yangtze River Delta region, China. Atmospheric Chemistry and Physics. 21**,** 2003-2025.

Cheng, J., et al., 2019. Dominant role of emission reduction in PM2.5 air quality improvement in Beijing during 2013–2017: a model-based decomposition analysis. Atmospheric Chemistry and Physics. 19**,** 6125-6146.

Ding, A., et al., 2019. Significant reduction of PM2.5 in eastern China due to regional-scale emission control: evidence from SORPES in 2011–2018. Atmospheric Chemistry and Physics. 19**,** 11791-11801.

Tong, D., et al., 2020. Dynamic projection of anthropogenic emissions in China: methodology and 2015–2050 emission pathways under a range of socio-economic, climate policy, and pollution control scenarios. Atmospheric Chemistry and Physics. 20**,** 5729-5757.

---

## Author Response (AR2)

**Manuscript ID: acp-2021-455**

**TITLE: Long-term trends and drivers of aerosol pH in eastern China**

We thank the editor and the reviewers for the comments and suggestions concerning our manuscript. They are valuable in helping us improve our manuscript. Below please find our point-by-pint responses to reviewers' comments.

**Comments of Reviewer #1:**

**Major concerns:**

This manuscript revision has addressed many of the concerns raised by the Referees in the first review. However, there are still some major issues with the manuscript that must be addressed before it can be published. Many of the issues are problems with the revised portions of the text. My specific concerns are:

1.   The results and discussion in Section 3.4 (Future Projections) need significant revision. Firstly, many prior studies have examined the sensitivity of PM2.5 to the precursors SO2, NOx, and NH3. None of those prior studies are cited or discussed. This work, as it relates to PM2.5 (and the individual aerosol species), is not novel. It is a much more simplified treatment of the topic than prior studies that use either a chemical transport model or a more rigorous thermodynamic analysis of the system, the typical approaches to this topic. The effect of the emissions scenarios on aerosol pH is quite novel and should be the focus of this section. Therefore, I recommend a substantial revision of this section (this comment also pertains to the Abstract and Conclusions where these results are discussed). As part of this revision, the limitations and uncertainties associated with their analysis should also be discussed. If the authors choose to keep the core analysis, i.e., presenting the partitioning ratios in addition to pH, then major revisions should include a more detailed analysis of the thermodynamic system under each emission scenario. See as an example Nenes et al. (2021) and Vasilakos et al. (2018). It should also include a thorough discussion of prior studies that have examined the SNA system and sensitivity of aerosol abundances to precursor emissions.

*Response:* We thank the reviewer for the comments. Following the reviewer's suggestion, we've modified section 3.4 focusing mainly on the influence of emission scenarios on aerosol pH. We first tested the sensitivity of aerosol abundances to precursor emissions with the historical data (Fig. R1 and R2), and then estimated the future scenario predictions based on the relationship between aerosol abundances and

precursor emissions (Fig. R3). The limitations and uncertainties associated with this analysis are also discussed. The detailed modifications are listed below.

**We've modified Section 3.4 into (Line 330-407):**

"*We first tested the sensitivity of aerosol abundances to precursor emissions with the historical data (Fig. S10), the emissions of Shanghai were obtained by the Multi-resolution Emission Inventory for China (MEIC, http://meicmodel.org/, last access: 15 January 2020). We found that the non-volatile sulfate concentrations generally correlated linearly with that of the SO2 emissions. For the volatile TNO3 and NHX, the correlations are less linear, likely due to the different deposition velocities of gases and particles (Nenes et al., 2021; Pye et al., 2020; Weber et al., 2016). The historical emission reductions have resulted in a moderate pH decrease (Figure 1), a moderate increase (0.2% per year) in the NO3- partitioning, and a decrease (-0.6% per year) in the NH4+ partitioning (Figure S11).*

[Figure]

Figure R1 *(added as Fig. S10 in the revised supplement)*. (a)-(c) Correlations of aerosol abundances to precursor emissions from 2011 to 2019, including (a)$SO_4^{2-}$ vs. $SO_2$, (b) $TNO_3$ vs. $NO_x$ and (c)NHx vs. $NH_3$. (d)-(f) Annual values of aerosol abundances and precursor emissions from 2011 to 2019, including (d)$SO_4^{2-}$ and $SO_2$ emission, (e)$TNO_3$, $NO_3^-$, and $NO_x$ emission, and (f) NHx, $NH_4^+$, and $NH_3$ emission.

[Figure]

Figure R2*(added as Fig. S11 in the revised supplement)*. Annual values of pH, $NO_3^-$ partitioning ($NO_3^-$ / ($NO_3^-$ + $HNO_3$)), and $NH_4^+$ partitioning ($NH_4^+$ / ($NH_4^+$ + $NH_3$)) from 2011 to 2019.

[revised manuscript text omitted]

**In addition, we've added Fig. R1 and R2 as Fig. S10 and S11 in the revised supplement.**

**Furthermore, we'd like to emphasize that this part is not the key point of this study. The main focus and novelty of our study are for the first time to explain the "Long-term trends and drivers of aerosol pH in eastern China". These main conclusions stand independent of future implications (Sect. 3.4). Thus, we are also open to remove Sect 3.4 and focus on the main content if the reviewers prefer with this option.**

**Technical/Minor Comments**

There are some minor, mostly technical corrections that should also be made:

- Line 73: change "particulate" to particles

- Line 113: delete "using meteorological parameters monitor"

- Line 169-170: suggest changing "the one-at-a-time method" to a more technical description

- Line 187: delete "obvious"

- Line 189: change "the implement" to "implementation"

- Line 194: change "Despite of the" to "Despite the"

- Line 198: change "with daily pH ranged" to "with a daily pH range"

- Line 199-200: I do not think Table S1 is necessary, especially because it is not discussed to compare and contrast the present results

- Line 201: it is not correct to call it a "pH level"

- Line 221: delete "the"

- Line 237-238: the sentence beginning "This is with similar seasonal…" needs to be revised for grammar

- Line 280-282: the sentence beginning "After sunrise, increase of temperature…" needs to be revised for grammar

- Line 288: change "roles" to "role"

- Section 3.4: "partition ratio" should be "partitioning ratio"

- Line 377: change "on" to "to"

- Line 380: suggest changing "revealed"

- Line 381: suggest adding a period after "respectively"

- Line 389: delete "existed"

- In addition to the above technical corrections, the entire manuscript should be carefully reviewed and edited for grammar, especially any changes made to the manuscript

  *Response:* We thank the reviewer for the comments and have corrected accordingly in the revised manuscript. In addition, we've read through the manuscript and did the grammar checks carefully.

**Comments of the Editor:**

The comments of reviewer 1 should be carefully followed. In the following, some comments from the editor, partly on similar points as Reviewer 1.

2.  The question of how organic carbon was measured is not answered satisfactorily. The time resolution is not given for example: was this also with 1 hour resolution as for the MARGA measurements? I doubt that this was the case. So it is important to mention in the manuscript that ALWCo is only a minor fraction of total ALWC. (Otherwise it would not be possible to determine reasonable diurnal variations). Also, make sure that all other relevant information given in the answers to the reviewers is also mentioned

in the manuscript.

*Response:* We thank the editor for the comments. The Thermal/Optical Carbon Aerosol Analyzer has a time resolution 1 hour, but the ALWCo was calculated using the annual average data. We've recalculated the ALWCo with the hourly data, and added the relevant information in the revised manuscript as follows:

*Modifications in manuscript:*

(1) Line 120-122: "*A Thermal/Optical Carbon Aerosol Analyzer (model RT-4, Sunset laboratory Inc.) equipped with a $PM_{2.5}$ cyclone was used for the organic carbon measurement at a time resolution of 1 hour.*"

(2) Line 143-147: "*The concentration of organic aerosol was estimated by multiplying the measured concentration of organic carbon by a factor of 1.6 (Turpin and Lim, 2001). The average concentrations of $ALWC_o$ and $ALWC_i$ in Shanghai from 2011 to 2019 were 4.1 ($\pm$10.2) and 32.6 ($\pm$52.5) $\mu g/m^3$, respectively. $ALWC_o$ only accounted for 11.1% of the total aerosol liquid water content.*"

3. I still find the figure captions in Fig. 1, 3 and 5 not clear enough. You might wish to list this as follows, or similar: a colored bar below the dashed line (decrease in pH) means an increase in the concentrations of sulfate, xx, and a decrease in yyy, and vice versa.

*Response:* We thank the editor for the comments. We've added more captions in Fig. 1, 3 and 5 in the revised manuscript as follows:

*Modifications in manuscript:*

[Figure]

Figure R4 *(revised Fig. 1 in the manuscript)*. (a) Long-term trends in aerosol pH during 2011–2019 in Shanghai. Gray dots and black lines represent the daily pH values and 30-day moving average pH values, respectively. (b) Contributions of individual factors to the ΔpH from 2011 to 2019. Here the coloured bar plots indicate the factors contributing to the ΔpH between two adjacent scenarios, e.g., 2011 to 2013. The stacked color bars below the dashed line represent the factors that had negative impacts on ΔpH, and the stacked color bars above the dashed line represent the factors that had positive impacts on ΔpH. The meanings of the abbreviations: RH, relative humidity; Temp, temperature; NVCs, non-volatile cations; $NH_x$, total ammonia; $TNO_3$, total nitrate; Oths, others.

[Figure]

Figure R5 *(revised Fig. 3 in the manuscript)*. Contributions of individual factors to the ΔpH across the four seasons. Here the bar plots indicate the factors contributing to the ΔpH between two adjacent seasons, e.g., spring (MAM) to summer (JJA). The stacked color bars below the dashed line represent the factors that had negative impacts on ΔpH and the stacked color bars above the dashed line represent the increase in ΔpH. The meanings of the abbreviations: RH, relative humidity; Temp, temperature; NVCs, non-volatile cations; NH$_x$, total ammonia; TNO$_3$, total nitrate; Oths, others.

[Figure]

Figure R6 *(revised Fig. 5 in the manuscript)*. Contributions of individual factors to the ΔpH between day and night. Here the bar plots indicate the factors contributing to the ΔpH between two adjacent hour periods, e.g., 0:00 to 6:00. The stacked color bars below the dashed line represent the factors that had negative impacts on ΔpH and the stacked color bars above the dashed line represent the increase in ΔpH. The meanings of the abbreviations: RH, relative humidity; Temp, temperature; NVCs, non-volatile cations; NH$_x$, total ammonia; TNO$_3$, total nitrate; Oths, others.

4.   I also agree with Reviewer 1 that the description in Section 3.4 is still not sufficient.

The revision should include a more detailed analysis of the thermodynamic system under each emission scenario. Explanations for changes in the partitioning ratios should be given in all cases. For example, an explanation needs to be given why for the moderate scenario the partitioning ratio decreases, while it increases for the BHE scenario. The explanation for the trend change in the nitrate partitioning after 2040 is not convincing (there is no sharp decrease in SO42- between 2040 and 2050). Also, the NVCs and ALWC have been shown to have a strong impact on the pH for the measurement period. Therefore, the assumed changes in NVCs and calculated ALWC for the scenarios need to be shown as well.

*Response:* Thanks for the comment. We've rewrote this section. Please see our response to comment #1.

5.    Finally, there are still a lot of English corrections needed. A (non-exhaustive) list of examples is given by Reviewer 1, sometimes the text is even not clear because of the wording. All "partition ratios" should be changed to "partitioning ratios", not only in the text, but also in Figure 6. As there are many more examples of required English corrections, the manuscript will definitely profit from a thorough English editing.

*Response:* We thank the editor for the comments. We've corrected the relevant wording and made a thorough English editing.

---

## Author Response (AR3)

**Manuscript ID: acp-2021-455**

**TITLE: Long-term trends and drivers of aerosol pH in eastern China**

We thank the editor and the reviewers for the comments and suggestions concerning our manuscript. They are valuable in helping us improve our manuscript. Below please find our point-by-pint responses to reviewers' comments.

**Comments of Reviewer #1:**

1.   Line 166: Authors should include the standard deviation in Figure 1a. Based on the daily pH values, the pH changes from 2011 to 2019 are neglected.

Response: We thank the reviewer for the comments. We've added the standard deviation of 30-day moving average pH in Figure 1a (shaded area).

*Modifications in manuscript:*

Line 223-233: "

[Figure]

Figure R1 *(revised Figure 1(a) in the manuscript)*. Long-term trends in aerosol pH during 2011–2019 in Shanghai.

Gray dots and black lines represent the daily pH values and 30-day moving average pH values, respectively.

Shaded areas mark the standard deviation of 30-day moving average pH values."

2.   Line 179: The average pH changed 0.24 in 9 years, I don't think it is convincing to be called "moderate change".

*Response:* We thank the reviewer for the comments. We revised this sentence into: "Despite the substantial change of aerosol abundance and composition, the aerosol pH only shows a minor change."

3.    Line 181: This sentence does not make sense, please refine it. What do you mean by "moderately acidic"? pH 1.15 is quite acidic.

*Response:* We thank the reviewer for the comments. Here the "moderately acidic" refer to the average pH levels, and not the lowest pH level of 1.15. We've changed this into: "The $PM_{2.5}$ in Shanghai was moderately acidic with a daily pH averaging 3.18 and ranging from 1.15 to 5.62". We consider such an average pH level as "moderately acidic" compared with the "quite acidic" regions like SE-US ranging 0~2 (Guo et al., 2015; Pye et al., 2018; Nah et al., 2018).

4.    Line 184: Can author be more specific with pH values? It would be helpful to list what pH value you are comparing to in different places?

*Response:* We've specified the pH values in the revised manuscript as follows (see Line 191-194): "Compared with other countries globally (Table S1), aerosol pH values in Chinese cities of 1.82 to 5.70 were higher than those in US cities of 0.55 to 2.20 (Guo et al., 2015; Pye et al., 2018; Nah et al., 2018), yet similar to those in European cities of 2.30 to 3.90 (Guo et al., 2018; Masiol et al., 2020)."

5.    Figure 1b: this plot is very confusing to read, what's the unit of each colored bar? They all look like not have the same length, how to compare them together?

*Response:* We thank the reviewer for the comments. To better show the factor contributions of the ΔpH between two adjacent scenarios, we've modified Figure 1b, 3 and 5 into waterfall plots. The waterfall plots show how an initial value (for example, the initial pH) is affected by a series of positive and negative values (https://support.microsoft.com/en-us/office/create-a-waterfall-chart-8de1ece4-ff21-4d37-acd7-546f5527f185; https://r-charts.com/flow/waterfall-chart/).

*Modifications in manuscript:*

(1)Line 223-233: "

[Figure]

Figure R2 *(revised Figure 1(b) in the manuscript)*. Contributions of individual factors to the ΔpH from 2011 to 2019. Here the black bars indicate the mean pH of different years, and the red and green bars represent the positive and negative effects of individual factors on ΔpH between two adjacent scenarios, e.g., 2011 to 2013, respectively. The meanings of the abbreviations: RH, relative humidity; Temp, temperature; NVCs, non-volatile cations; $NH_x$, total ammonia; $TNO_3$, total nitrate; Oths, others."

(2)Line 264-273: "

[Figure]

Figure R3(*revised Figure 3 in the manuscript*). Contributions of individual factors to the ΔpH across the four seasons. Here the black bars indicate the mean pH of different seasons, and the red and green bars represent the positive and negative effects of individual factors on ΔpH between two adjacent scenarios, e.g., spring (MAM) to summer (JJA), respectively. The meanings of the abbreviations: RH, relative humidity; Temp, temperature; NVCs, non-volatile cations; $NH_x$, total ammonia; $TNO_3$, total nitrate; Oths, others."

(3)Line 302-309: "

[Figure]

Figure R4(*Revised Figure 5 in the manuscript*). Contributions of individual factors to the ΔpH between day and night. Here the black bars indicate the mean pH of different hours, and the red and green bars represent the positive and negative effects of individual factors on ΔpH between two adjacent scenarios, e.g., 0:00 to 6:00, respectively. The meanings of the abbreviations: RH, relative humidity; Temp, temperature; NVCs, non-volatile cations; $NH_x$, total ammonia; $TNO_3$, total nitrate; Oths, others."

6. Line 235: Based on Figure 3, the temperature has a positive impact on summer and fall (JJA-SON), but a negative impact on spring and winter. Can author explain more about how the temperature changes the pKa and how much changes in temperature in those seasons?

*Response:* As explained in the caption of Fig. 3, the color bars represent the ΔpH between two adjacent seasons, e.g., spring to summer. The change of temperature, ΔT, is 11.8, -8.4, -13.3 and 10.0 °C during 4 scenarios, respectively, including spring to summer, summer to fall, fall to winter, and winter to spring. This will cause the $pK_a^*$ to change by -0.59, 0.41, 0.72 and -0.54, respectively (see detailed calculation method in Zheng et al. 2020), which are roughly in line with ΔpH levels.

7. Figure 5, same as an earlier comment, the colored bars is very confusing with different length and not easy to compare.

*Response:* Please see our response to comments #5.

8. Line 274: In Figure 4, it seems like RH has a similar trend with pH, especially at midnight and noon time. Do you think that the RH will be a factor that drives the change of pH, as lower RH (less water) leads to higher pH? Can author provide some discussion?

*Response:* To clarify that, Figure 4 shows that lower RH (less water) corresponds to lower pH, and not higher pH. Yes, RH is a factor driving the change of aerosol pH. According to the multiphase buffer theory (Zheng et al. 2020), lower ALWC leads to lower pH under an $NH_4^+/NH_3$ buffer system. Because the diurnal variation of $PM_{2.5}$ is rather weak (varying between 46~50 μg m$^{-3}$), the change of ALWC is mainly driven by the change of RH and lower ALWC corresponds to lower RH. We modified the ALWC axis in Fig. 4 to make the trend more clearly displayed. Please see the following changes.

*Modifications in manuscript:*

Line 296-299: "

[Figure]

Figure R5 *(revised Figure 4 in the manuscript)* Diurnal variations of the mass concentrations of major ions in $PM_{2.5}$, relative humidity (RH), temperature (Temp), predicted aerosol liquid water content (ALWC) and aerosol pH during 2011–2019 in Shanghai."

**Comments of Reviewer #2:**

1. One of the main concerns for the paper is the approach to assign the delta(pH) into

the contribution from sulfate, ammonium, ect… I do not quite follow the physical meaning of this calculation. It feels rather like a statistical approach to me. If we want pH1-pH2=pH(sulfate1)-pH(sulfate2) + pH(ammonium1)-pH(ammonium2) … then it needs to have linear relationship to all these factors? In reality, the pH response to these factors can be quite non-linear and have very different sensitivity in different regions. The authors try to address this issue with an additional "others" term, but the physical meaning of this term is doubtful, it can be the non-linear response not considered by the previous calculation treatment. In the Figure S3 of a separate study (Revisiting the Key Driving Processes of the Decadal Trend of Aerosol Acidity in the U.S, 10.1021/acsenvironau.1c00055), it looks ISORROPIA also has some non-smooth response to the parameters. Will such behavior affect the accuracy of this calculation approach?

*Response:* Indeed, the response can be nonlinear. Nevertheless, this one-at-a-time method (OAT) is a common sensitivity analysis method for evaluating the impact of changing one factor at a time in turn (Saltelli A, et al. 2000; Saltelli A, et al., 2007). Many early studies have adopted the OAT analysis to characterize the sensitivity of pH to different factors (Ding et al., 2019; Wang et al., 2020; Tao and Murphy, 2019). We've clarified this in the revised manuscript as (see Line 167-168): "Note that because of the nonlinear dependence of pH to different factors, the sum of contributions of individual factors can be slightly different from the overall contributions of all factors."

The reviewer was referring to cases of SE-US in summer 2008 (Fig. S3 of Zheng et al., 2022) as shown below. We've examined our case in Shanghai, and find no abrupt changes.

2. The dataset did not include gaseous HCl but treat the particulate Cl as overall Cl, while ammonium chloride formation can be a significant contributor to haze formation, at least in winter. Can the authors perform a sensitivity test for this assumption?

*Response:* We thank the reviewer for the comments. As shown in Fig. R6, we have performed a sensitivity test and compared the differences in pH calculated from simulations with and without gaseous HCl. Because of the relatively low abundance of gaseous HCl, the overall Cl (HCl + Cl$^-$) shows a minor change resulting in little difference in pH between two simulations.

[Figure]

Figure R6 The differences in pH calculated from simulations with and without gaseous HCl from 2011 to 2019.

[Figure]

Figure R7 The box plots of HCl and Cl⁻ concentrations from 2011 to 2019.

3. Section 3.1.1 & 3.1.2, it's better for the authors to present the annual trends of meteorological parameters prior to chemical composition changes since temperature and RH are also influencing pH values.

*Response:* We thank the reviewer for this constructive suggestion. We've added the related analysis on trends of temperature and RH in the revised manuscript and supplement. Details are provided as follows:

*Modifications in manuscript:*

Line 114-117: "Temperature and RH, which are important factors affecting aerosol pH, were also measured at a time resolution of 1 min. Annually averaged temperature and RH from 2011 to 2019 are shown in Figure S6. The t-test results revealed that temperature rose significantly at a rate of 1.2 %/yr ($p < 0.01$), while RH changed little."

*Modifications in* **supplement***:*

We've added the Figures S6 in the revised supplement as: "

[Figure]

Figure R8 *(added Figure 6 in the manuscript)* Annual values of temperature (T) and relative humidity (RH) from 2011 to 2019"

**Some other comments:**

4.  Line 29-30: It's better to address which approach is used to get these numbers in the abstract.

*Response:* Thanks for the suggestion. We added the description of the approach in the abstract as follows:

*Modifications in manuscript:*

Line 23-24: "Here, we reported the first trend analysis of aerosol pH from 2011 to 2019 in eastern China, calculated with ISORROPIA model based on observed gas and aerosol compositions."

5.  Line 226-227: I don't understand how concentrations of aerosol chemical composition impact pH. Is there a relationship such as lower PM days having higher

pH values?

*Response:* We thank the reviewer for the comments. This is well explained in Zheng et al. 2020, that at given RH and temperatures, the lower aerosol concentrations will lead to lower ALWC at similar chemical composition characteristics, which will lead to lower pH.

6. Line 35: formation in the gas phase?

*Response:* Thanks for the comments. We revised this sentence into: "The corresponding aerosol pH in eastern China is estimated to increase by ~0.19, resulting in 4% more $NO_3^-$ and 12% more $NH_4^+$ partitioning in the gas phase,"

7. Line 102: equivalent concentrations of cation and anion? Should be more specific.

*Response:* Thanks for the comments. Here the equivalent concentrations mean charge equivalent concentrations. We've further clarified this point as (see Line 102-104): "To ensure the data quality, the ion balance between the measured charge equivalent concentrations of cation ($NH_4^+$, $Na^+$, $K^+$, $Ca^{2+}$ and $Mg^{2+}$) and anion ($SO_4^{2-}$, $NO_3^-$ and $Cl^-$) species was examined as shown in Figure S2."

**Comments of Editor:**

I note the substantial improvements based on the reviewer comments.

Still, further improvements are required; besides the ones of the reviewers I note the following ones (all line numbers refer to the file 455-ATC2):

8. L333: the statement 'the sulfate concentrations generally correlated linearly with that of the SO2 emissions' is not correct: Fig. S10 shows that the SO2 emission decreases by a factor of 4, while the sulfate concentration decreased by less than a factor of 2. This is not linear, so the authors should rephrase this and give an explanation why these two variable do not behave linearly.

*Response:* Thanks for the comments. As shown in Fig. S10, the sulfate concentrations are generally linear (following the relationship of $y = kx + b$) rather than proportional (following the relationship of $y = kx$) to SO$_2$ emissions. The sulfate concentrations and SO$_2$ emissions are reduced by different proportions due to the presence of the intercept. The intercept indicates that, even if the anthropogenic SO$_2$ emissions are reduced to zero, there will still be background SO$_4^{2-}$ aerosol from natural sources and regional transport.

9. I also note that there are drastic differences in Fig. 6, e.g, for pH in the BHE scenario (6d) between the previous and the current version. What are the reasons for these differences? And are the authors convinced that these numbers are now the correct ones?

*Response:* The difference is due to the change of scenario projections of sulfate concentrations. Instead of assuming sulfate concentration is proportional to $SO_2$ emissions (previous version, we now adopt a linear dependence, i.e., aerosol concentrations $= k *$ (precursor emissions) $+ b$. We consider the current projection more robust as they're supported by the historical trends.

10. I also appreciate the substantial improvements in the text. Still, this manuscript will profit from a further thorough English editing; the following points are just examples:

L25: Yangtze: add 'the'

L28: units: not needed. If you want to use it, it should read units (make it consistent throughout the manuscript)

L36: formation in the gas phase: not clear, as neither NO3- nor NH4+ is formed in the gas phase.

L45: solubility(Cheng: please make sure to add required space before references here and in all future instances (many).

L86: providing scientific: add 'a'

L93: this sampling site represent: add 's'

L104: thus is: replace by 'are'

L107: Lithium Bromide: no capitals

L108: the multi-points calibrations: replace by multi-point calibrations

L110: ion balance: add 'the'

L113: R2: all symbols should be italic; between the cation and anion: add 's' to both

L161: are not well corrected: do you mean correlated?

L163: be attribute: add 'd'

L165: are high uncertainty: wrong English; correct

L171: differs with: replace by differs from

L186: 0.04 unit pH per year: replace by 0.04 pH units per year

L190: before the implement of the Action Plan: implementation

L194:were kept being: replace by remained

L203: aerosol pH: add 'values'

L226: from 2013 to 2019, respectively: replace by , respectively from 2013 to 2019

L236: implementation of Action Plan: add 'the'

L252: similar seasons: replace by the

L287: were depicted: replace by are depicted

L288: effects individual: add 'of'

L289: Bar plot: add 'the'

L295: max: replace by 'maximum'

L297: with ALWC reached: replace by.. 'reaching'

L330: reference scenario that without: delete 'that'

L346: and 2.6 μg/m3: add ', respectively

L351: NH4+ partitioning ratio: add 'the'

L362: indicating that more ammonium will exist in the gas phase as NH3: replace by ..relatively more ammonium.. (as NH3 emission and concentration will be reduced)

L378: are: replace by is

L381: are: replace by is

L413: in YRD region: add 'the'

*Response:* We thank the reviewer for the comments and have corrected accordingly in the revised manuscript. In addition, we've read through the manuscript and did the grammar checks carefully.

Reference

Ding, J., Zhao, P., Su, J., Dong, Q., Du, X., and Zhang, Y.: Aerosol pH and its driving factors in Beijing, Atmospheric Chemistry and Physics, 19, 7939-7954, 10.5194/acp-19-7939-2019, 2019.

Guo, H., Xu, L., Bougiatioti, A., Cerully, K. M., Capps, S. L., Hite, J. R., Carlton, A. G., Lee, S. H., Bergin, M. H., Ng, N. L., Nenes, A., and Weber, R. J.: Fine-particle water and pH in the southeastern United States, Atmospheric Chemistry and Physics, 15, 5211-5228, 10.5194/acp-15-5211-2015, 2015.

Guo, H., Otjes, R., Schlag, P., Kiendler-Scharr, A., Nenes, A., and Weber, R. J.: Effectiveness of

ammonia reduction on control of fine particle nitrate, Atmospheric Chemistry and Physics, 18, 12241-12256, 10.5194/acp-18-12241-2018, 2018.

Masiol, M., Squizzato, S., Formenton, G., Khan, M. B., Hopke, P. K., Nenes, A., Pandis, S. N., Tositti, L., Benetello, F., Visin, F., and Pavoni, B.: Hybrid multiple-site mass closure and source apportionment of PM2.5 and aerosol acidity at major cities in the Po Valley, Sci Total Environ, 704, 135287, 10.1016/j.scitotenv.2019.135287, 2020.

Nah, T., Guo, H., Sullivan, A. P., Chen, Y., Tanner, D. J., Nenes, A., Russell, A., Ng, N. L., Huey, L. G., and Weber, R. J.: Characterization of aerosol composition, aerosol acidity, and organic acid partitioning at an agriculturally intensive rural southeastern US site, Atmospheric Chemistry and Physics, 18, 11471-11491, 10.5194/acp-18-11471-2018, 2018.

Pye, H. O. T., Zuend, A., Fry, J. L., Isaacman-VanWertz, G., Capps, S. L., Appel, K. W., Foroutan, H., Xu, L., Ng, N. L., and Goldstein, A. H.: Coupling of organic and inorganic aerosol systems and the effect on gas-particle partitioning in the southeastern US, Atmos Chem Phys, 18, 357-370, 10.5194/acp-18-357-2018, 2018.

Saltelli, A., Chan, K., Scott, E.M. Sensitivity Analysis, John Wiley & Sons, New York. 2000.

Saltelli, A. Ratto, M., Andres, T., Campolongo, F., Cariboni, J. Gatelli, D., Saisana, M., Tarantola, S. Global Sensitivity Analysis. The Primer, John Wiley & Sons, New York. 2007.

Tao, Y. and Murphy, J. G.: The sensitivity of PM2.5 acidity to meteorological parameters and chemical composition changes: 10-year records from six Canadian monitoring sites, Atmospheric Chemistry and Physics, 19, 9309-9320, 10.5194/acp-19-9309-2019, 2019.

Wang, S., Wang, L., Li, Y., Wang, C., Wang, W., Yin, S., and Zhang, R.: Effect of ammonia on fine-particle pH in agricultural regions of China: comparison between urban and rural sites, Atmospheric Chemistry and Physics, 20, 2719-2734, 10.5194/acp-20-2719-2020, 2020.

Zheng, G., Su, H., Wang, S., Andreae, M. O., Poschl, U., and Cheng, Y.: Multiphase buffer theory explains contrasts in atmospheric aerosol acidity, Science 369, 1374-1377, 2020.

Zheng, G., Su, H., and Cheng, Y.: Revisiting the Key Driving Processes of the Decadal Trend of Aerosol Acidity in the U.S, ACS Environmental Au, 10.1021/acsenvironau.1c00055, 2022.